# Control of anterior GRadient 2 (AGR2) dimerization links endoplasmic reticulum proteostasis to inflammation

Marion Maurel[1,2,3,4,†], Joanna Obacz[1,2,†], Tony Avril[1,2,†], Yong-Ping Ding[5,6,7], Olga Papadodima[8], Xavier Treton[5,6,7], Fanny Daniel[5,6,7], Eleftherios Pilalis[8,9], Johanna Hörberg[10], Wenyang Hou[11], Marie-Claude Beauchamp[11], Julien Tourneur-Marsille[5,6,7], Dominique Cazals-Hatem[5,6,7], Lucia Sommerova[12], Afshin Samali[4] ID, Jan Tavernier[3], Roman Hrstka[12], Aurélien Dupont[13], Delphine Fessart[14], Frédéric Delom[14], Martin E Fernandez-Zapico[15], Gregor Jansen[16] ID, Leif A Eriksson[10], David Y Thomas[16], Loydie Jerome-Majewska[11], Ted Hupp[9,12,17], Aristotelis Chatziioannou[8,18,*] ID, Eric Chevet[1,2,**] ID & Eric Ogier-Denis[5,6,7,***] ID

## Abstract

Anterior gradient 2 (AGR2) is a dimeric protein disulfide isomerase family member involved in the regulation of protein quality control in the endoplasmic reticulum (ER). Mouse AGR2 deletion increases intestinal inflammation and promotes the development of inflammatory bowel disease (IBD). Although these biological effects are well established, the underlying molecular mechanisms of AGR2 function toward inflammation remain poorly defined. Here, using a protein–protein interaction screen to identify cellular regulators of AGR2 dimerization, we unveiled specific enhancers, including TMED2, and inhibitors of AGR2 dimerization, that control AGR2 functions. We demonstrate that modulation of AGR2 dimer formation, whether enhancing or inhibiting the process, yields pro-inflammatory phenotypes, through either autophagy-dependent processes or secretion of AGR2, respectively. We also demonstrate that in IBD and specifically in Crohn's disease, the levels of AGR2 dimerization modulators are selectively deregulated, and this correlates with severity of disease. Our study demonstrates that AGR2 dimers act as sensors of ER homeostasis which are disrupted upon ER stress and promote the secretion of AGR2 monomers. The latter might represent systemic alarm signals for pro-inflammatory responses.

**Keywords** AGR2; endoplasmic reticulum; inflammation; proteostasis; TMED2
**Subject Categories** Digestive System; Immunology; Metabolism

1  INSERM U1242, "Chemistry, Oncogenesis Stress Signaling", University of Rennes, Rennes, France
2  Centre de Lutte Contre le Cancer Eugène Marquis, Rennes, France
3  VIB Department of Medical Protein Research, UGent, Gent, Belgium
4  Apoptosis Research Centre, School of Natural Sciences, NUI Galway, Galway, Ireland
5  INSERM, UMR1149, Team «Gut Inflammation», Research Centre of Inflammation, Paris, France
6  Université Paris-Diderot Sorbonne Paris-Cité, Paris, France
7  APHP Beaujon Hospital Clichy la Garenne, Paris, France
8  Institute of Biology, Medicinal Chemistry & Biotechnology, NHRF, Athens, Greece
9  International Centre for Cancer Vaccine Science, Gdansk, Poland
10  Department of Chemistry and Molecular Biology, University of Gothenburg, Göteborg, Sweden
11  Departments of Anatomy and Cell Biology, Human Genetics, and Pediatrics, McGill University, Montreal, QC, Canada
12  Regional Centre for Applied Molecular Oncology (RECAMO), Brno, Czech Republic
13  Microscopy Rennes Imaging Centre, and Biosit, UMS3480 CNRS, University of Rennes 1, Rennes Cédex, France
14  University of Bordeaux, Bordeaux, France
15  Division of Oncology Research, Department of Oncology, Schulze Center for Novel Therapeutics, Mayo Clinic, Rochester, MN, USA
16  Biochemistry Department, McGill University Life Sciences Complex, Montréal, QC, Canada
17  Edinburgh Cancer Research Centre at the Institute of Genetics and Molecular Medicine, Edinburgh University, Edimburgh, UK
18  e-NIOS PC, Kallithea-Athens, Greece
   *Corresponding author. Tel: +30 2107273751; E-mail: achatzi@eie.gr
   **Corresponding author. Tel: +33 223237258; E-mail: eric.chevet@inserm.fr
   ***Corresponding author. Tel: +33 157277307; E-mail: eric.ogier-denis@inserm.fr
   †These authors contributed equally to this work

# Introduction

The regulation of protein homeostasis (proteostasis) in the endoplasmic reticulum (ER) has recently emerged as an important pathophysiological mechanism involved in the development of different diseases (Hetz *et al*, 2015). The capacity of the ER to cope with the protein misfolding burden is controlled by the kinetics and thermodynamics of folding and misfolding (also called proteostasis boundary), which are themselves linked to the ER proteostasis network capacity (Powers *et al*, 2009). The ER ensures proper folding of newly synthesized proteins through the coordinated action of ER-resident molecular chaperones, folding catalysts, quality control, and degradation mechanisms. Anterior gradient 2 (AGR2), a folding catalyst, binds to nascent protein chains, and it is required for the maintenance of ER homeostasis (Persson *et al*, 2005; Higa *et al*, 2011; Chevet *et al*, 2013). Loss of AGR2 has been associated with intestinal inflammation (Park *et al*, 2009; Zhao *et al*, 2010), and several studies have demonstrated that unresolved ER stress leads to spontaneous intestinal inflammation (Kaser *et al*, 2013). Although anterior gradient proteins, including AGR2, were identified more than a decade ago, their precise biological functions remain ill-defined. AGR2 was first defined as an ER-resident foldase (Chevet *et al*, 2013) and was also shown to exhibit extracellular activities (Fessart *et al*, 2016), and all of these functions most likely depend on protein–protein interactions. Previously, a few AGR2 interacting partners have been identified (Maslon *et al*, 2010; Yu *et al*, 2013) and fewer have been validated as genuine AGR2 binding partners in the ER. Moreover, biochemical approaches have shown that AGR2 forms dimers (Ryu *et al*, 2012; Patel *et al*, 2013). As such, this justifies an in-depth study to characterize the protein–protein interaction-dependent regulatory mechanisms controlling AGR2 dimerization and functions.

In mammals, AGR2 is generally present in mucus secreting epithelial cells and is highly expressed in Paneth and goblet intestinal cells, with the highest levels in the ileum and colon (Komiya *et al*, 1999; Chang *et al*, 2008a,b). In goblet cells, AGR2 forms mixed disulfide bonds with Mucin 2 (MUC2), allowing for its correct folding and secretion (Park *et al*, 2009; Zhao *et al*, 2010). MUC2 is an essential component of the gastrointestinal mucus covering the epithelial surface of gastrointestinal tract to confer the first line of defense against commensal bacteria. Knockout of AGR2 inhibits MUC2 secretion by intestinal cells, thereby decreasing the amount of intestinal mucus and leading to a spontaneous granulomatous ileocolitis, closely resembling human inflammatory bowel disease (IBD; Zhao *et al*, 2010). Accordingly, decreased AGR2 expression and some of AGR2 variants were identified as risk factors in IBD (Zheng *et al*, 2006). However, despite the strong link between AGR2 and the etiology of IBD, the molecular mechanism by which AGR2 regulates its activity and contributes to the development of IBD still remains elusive. To unveil the molecular functions of AGR2 and its pathophysiological roles, we have developed the ER Mammalian protein–protein Interaction Trap (ERMIT), which allows us to specifically detect AGR2 protein–protein interactions in the ER. We found that upon ER proteostasis alteration AGR2 dimer was disrupted and a siRNA screen using ERMIT identified TMED2 as a major regulator of AGR2 dimerization. Moreover, we demonstrated that variations in TMED2 expression resulted in AGR2 secretion and pro-inflammatory phenotypes *in vitro*, in mouse models and in patients' samples. Hence, we propose that ER proteostasis-mediated control of AGR2 dimerization, which might depend on TMED2, promotes AGR2 release in the extracellular environment thereby enhancing monocyte recruitment and pro-inflammatory phenotypes.

# Results

### AGR2 forms stress-regulated homodimer in the endoplasmic reticulum

Structural studies showed that AGR2 forms dimers through residues E60 (Fig 1A) and C81 (Ryu *et al*, 2012; Patel *et al*, 2013), respectively. Results involving E60 in AGR2 dimerization were confirmed using molecular dynamics (Fig 1B). The dimeric versus monomeric equilibrium of AGR2 was also investigated using molecular

---

**Figure 1.  AGR2 dimerization with ERMIT assay, principles, and validation of the method.**

The ERMIT assay relies on the signaling properties of IRE1, one of the three ER stress sensors and reports for a dimerization event occurring in the lumen of the ER. The assay can be applied to heterodimerization or homodimerization events.

A    Upper panel: NMR structure of the non-covalent dimer of AGR2 (PDB ID: 2LNS). The dimer domain is highlighted with a yellow circle (residue 54–70 of each monomer). Monomer A is colored green and monomer B in pink. The figure was generated in Chimera. Lower panel: a close-up illustration of the dimer domain, showing that the dimer is stabilized through two salt bridges between E60 and K64 of each monomer.

B    Molecular modeling showing root-mean-square deviation (RMSD) plot for the MD simulation of the AGR2 WT and E60A mutant. The interaction energies are defined as the sum of the short-range Coulomb interactions and the short-range Lennard-Jones potential between monomer A and B.

C, D  Western blot showing the expression of AGR2 dimers (D) and monomers (M) in HEK293T subjected to DSP-mediated cross-linking and that were previously transfected with either a control siRNA (siCTL) or a siRNA targeting AGR2 (siAGR2) for 24 h (C) or treated or not with tunicamycin (Tun) prior to cross-linking (D). Reduced or non-reduced samples were resolved by SDS–PAGE and immunoblotted using anti-AGR2, anti-ERK1, or anti-calnexin (CANX) antibodies (for loading control).

E    Principles of the ERMIT assay. A wild-type IRE1 bait is used to report for dimerization, whereas a kinase catalytic mutant (K599A) is used as a control to prove that signal observed with the wild-type form is due to IRE1 activation.

F    AGR2 dimerization was monitored with ERMIT [wild-type (WT)]. The luminescence signal was abrogated when using the kinase dead (KD) constructs and by the constructs exhibiting mutations in the dimerization domain [E60A or C81S or double mutant (DM)]. The graph represents average signal normalized on reporter protein expression $\pm$ SD [$n = 5$; **: WT/KD ($P = 0.002$), E60/WT ($P = 0.0034$), E60/KD ($P = 0.001$), C81/KD ($P = 0.004$), DM/WT ($P = 0.0021$), and DM/KD ($P = 0.0017$), respectively; *: C81/WT ($P = 0.0098$)]. The Mann–Whitney statistical test was used.

G    Cells expressing various bait and prey constructs and the XBP1 splicing reporter or the XBP1 splicing reporter alone were exposed to increasing concentrations of DTT. The ERMIT signals obtained with both baits were then normalized to that of XBP1s to obtain results independent of the activation of endogenous IRE1 by the use of chemical ER stressors ($n = 4$). The graph represents average signal normalized on reporter protein expression $\pm$SD.

---

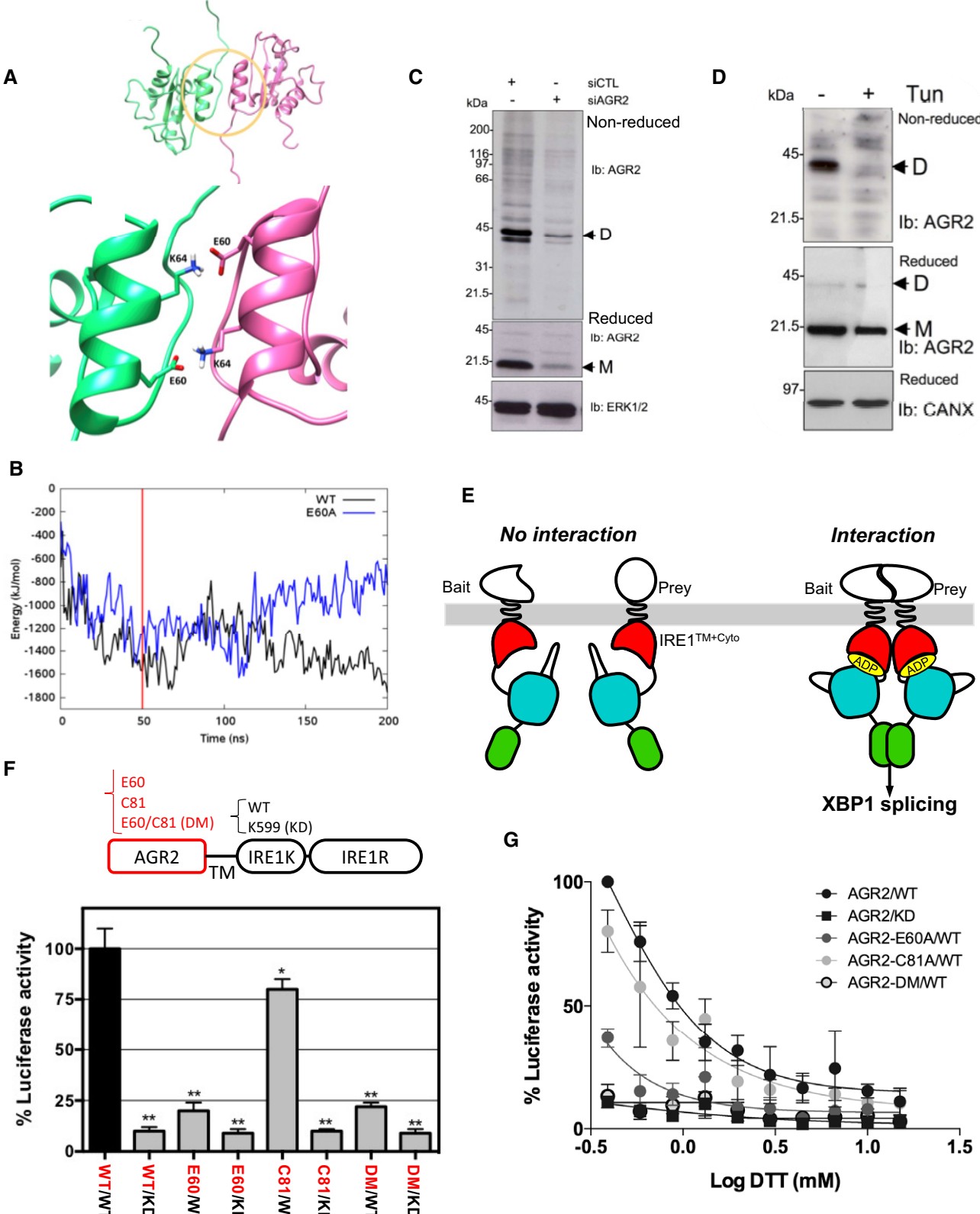

**Figure 1.**

modeling approaches. Indeed, the reduced dimer stability of the E60A mutant was verified by performing 200 ns molecular dynamics simulations of wild-type and mutant dimers (Appendix Fig S1A–F). E60 of each monomer stabilizes the dimer by forming salt bridges to K64 of the other monomer. The WT system remains stable throughout the simulation, whereas the E60A mutant form rapidly dissociates, as identified in increased RMSD, radius of gyration, and distance measurements, concomitant with loss of interaction energy. These results indicate that AGR2 might exist under both monomeric and homodimeric forms.

To validate the dimerization of AGR2 in our cellular models, cells were transfected with a previously validated siRNA against AGR2 (Higa et al, 2011) and its corresponding control siRNA. Cells were then treated with the chemical cross-linker DSP. Cross-linked proteins were resolved on either non-reducing (top blot; Fig 1C) or reducing (middle blot; Fig 1C) conditions and analyzed by Western blot. Data included in Fig 1C revealed that AGR2 exists predominantly as homodimers. Since AGR2 is also involved in protein quality control in the ER (Higa et al, 2011), we evaluated the impact of ER homeostasis disruption on AGR2 dimerization. DSP-mediated protein cross-linking of tunicamycin-treated cells revealed that AGR2 homodimers disappeared upon ER stress induced by tunicamycin, whereas total AGR2 expression levels did not change significantly (Fig 1D).

To further dissect the mechanisms by which AGR2 dimerizes, we developed the ERMIT assay (Fig 1E). ERMIT is a mammalian two-hybrid method, adapted from the existing ER-MYTH yeast assay (Jansen et al, 2012) and based on the functional complementation of the IRE1 signaling pathway. IRE1 is normally maintained in an inactive state by its association with the molecular chaperone BiP. Upon accumulation of misfolded proteins in the ER, initiating ER stress, IRE1 competes with those proteins for binding to BiP. When activated, IRE1 cleaves XBP1 mRNA at two consensus sites to initiate an unconventional splicing reaction. This spliced mRNA leads to the generation of a functional XBP1 transcription factor (Hetz et al, 2015). In the ERMIT assay, the luminal domain of IRE1 was replaced by different bait proteins (Fig 1E), and independently of ER stress, bait and prey interactions lead to IRE1 activation and subsequent XBP1 splicing. This splicing is monitored by a XBP1 splicing luciferase reporter system (Hetz et al, 2015).

To determine whether AGR2 dimerizes in the ER, we replaced the luminal domain of IRE1 with AGR2 wild-type (WT), or two AGR2 dimerization inactive mutants [E60A, C81A, or the E60A/C81A double mutant (DM)]. The transmembrane and WT or kinase dead (KD) cytosolic domains of IRE1 were used as positive controls. These AGR2-IRE1 chimeric constructs were transfected into HEK293T cells, and their expression and localization to the ER were verified by Western blot (Appendix Fig S1G) and immunofluorescence microscopy (Appendix Fig S1H). ERMIT signals produced by HEK293T cells transfected with the different AGR2 baits were then quantified (Fig 1F). As IRE1 overexpression induces its auto-activation (Hetz et al, 2015), the ERMIT assay was optimized using low quantities of the transfected plasmids to ensure that no IRE1 auto-activation was detectable. In confirmation of the validity of the activation assay, all the IRE1 KD baits reduced the luminescence signal by more than 90% (Fig 1F), thus confirming that the signal observed was not due to the activation of endogenous IRE1. The AGR2-WT bait produced the highest signal indicating that the

dimerization of AGR2 occurred in the ER. The C81A mutant showed a 25% decrease in the signal, relative to AGR2-WT, whereas the E60A or the DM reduced the signal by about 80%. This demonstrates that AGR2 dimerizes in the ER and that the E60 residue plays a key role in this in vivo interaction whereas the C81 does not. Moreover, ER stress induced by DTT treatment showed a dose-dependent dissociation of AGR2 homodimers as assessed by the decrease in luminescence observed for all the constructs tested (Fig 1G). The same result was observed when ER stress was induced by thapsigargin or tunicamycin (Appendix Fig S1I). An IC50 was then calculated for each of the ER stressors (Appendix Fig S1J).

Stress-related AGR2 functions in the ER were also evaluated using $^{35}$S-methionine pulse-chase followed by AGR2 immunoprecipitation to investigate the dynamics of AGR2 binding to other partners. Five AGR2 binding partners were visualized using this method in HeLa cells (bands 1–5, Appendix Fig S2A). Interestingly, the kinetics of association of these proteins with AGR2 differed between basal and ER stress conditions. The association of the proteins corresponding to bands 2, 3, and 4 with AGR2 was destabilized upon ER stress, while that of proteins corresponding to bands 1 and 5 was stabilized (Appendix Fig S2A). These data led us to propose a model in which AGR2 exists mainly as a homodimer when protein-folding demand does not overwhelm the cellular folding capacity but in case of stress, AGR2 homodimers dissociate to unveil their chaperone/quality control properties. Moreover, our data also suggest that the ratio of monomeric versus dimeric AGR2 might represent a potent mean to selectively control ER proteostasis.

## Identification of AGR2 dimer regulators and functional characterization of TMED2

To characterize the mechanisms regulating AGR2 dimeric versus monomeric status, we designed a specific ERMIT-based siRNA screen and tested the impact of a custom-designed siRNA library that targets 274 ER-resident proteins (Fig 2A). The counterscreen used cells transfected only with the XBP1s reporter (Fig 2A). We identified siRNAs that are positively or negatively modulating AGR2 dimer formation and allowed the identification of proteins that act as either inhibitors or enhancers of dimerization. A total of 71 proteins representing candidate AGR2 homodimer enhancers (42) or inhibitors (29) were identified (Fig 2B and Appendix Table S1). Functional pathway analysis based on Gene Ontology and Reactome annotations of these candidates revealed an enrichment of AGR2 homodimer enhancers in protein productive folding and ERAD processes, while AGR2 homodimer inhibitors were significantly enriched in functions related to calcium homeostasis, ER stress, and cell death processes (Fig 2C and Appendix Fig S2B–D). Remarkably, a high network connectivity was observed between dimer enhancers (green, Fig 2D) or inhibitors (red, Fig 2E), thereby confirming AGR2 functions in productive protein folding when dimeric and managing misfolded proteins (stress) when monomeric. These data also confirm our primary hypothesis and reinforce the importance of AGR2 dimerization control in proper functioning of the ER.

Among the positive regulators of AGR2 dimerization found in the screen (Fig 2), we identified TMED2, a p24 family member previously shown to function as a cargo receptor (Barlowe, 1998). Moreover, p24 family members in the yeast Saccharomyces cerevisiae

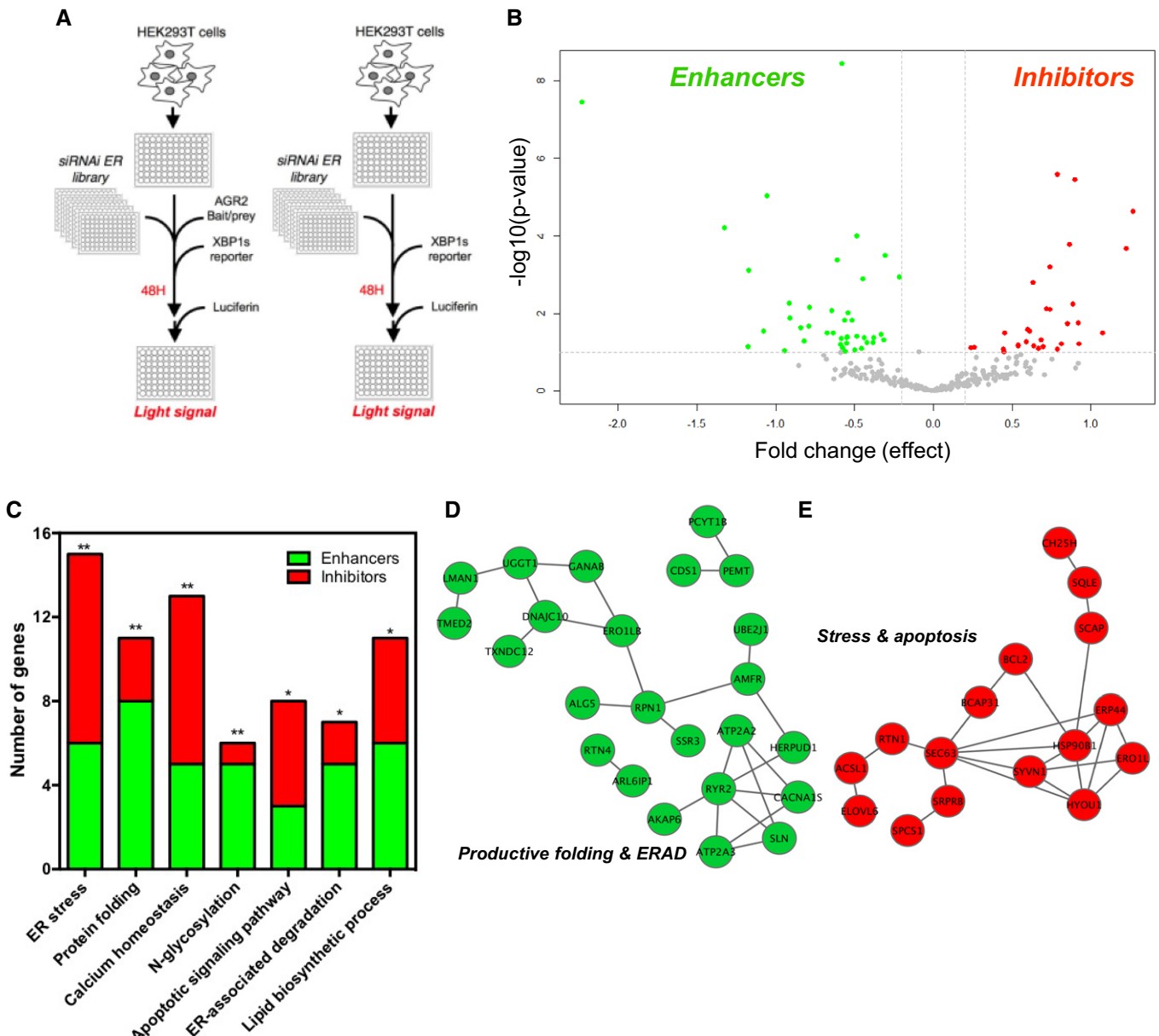

**Figure 2. siRNA screen for identifying regulators of AGR2 dimer formation.**

A The first screen was carried out using HEK293T cells transfected with the ERMIT bait/prey, system, the XBP1 splicing reporter, and the siRNA library (*n* = 5). The counter screen was carried out using the same experimental setup without the ERMIT bait/prey system (*n* = 4).

B Volcano plot representing the statistical analysis of the data from the screen, where the *y* axis represents statistical criteria and the *x* axis the fold change in luminescence units.

C Number of genes associated with the indicated processes, as revealed by functional analysis based on Gene Ontology and Reactome terms, (*): Apoptotic signaling pathway (*P* = 0.0083), ER-associated degradation (*P* = 0.0076), and lipid biosynthetic process (*P* = 0.0059), respectively; (**): ER stress (*P* = 0.0044), Protein folding (*P* = 0.003), Calcium homeostasis (*P* = 0.0022), N-glycosylation (*P* = 0.0018), respectively.

D, E Enrichment of AGR2 homodimer enhancers in functions associated with ER homeostasis (green) and of AGR2 homodimer inhibitors in functions associated with ER homeostasis imbalance (red).

were shown to interact with PDI, the family of proteins to which AGR2 belongs (Appendix Table S2). To further characterize the functional interaction between TMED2 and AGR2, we first evaluated whether these 2 proteins could be found in a complex. As such, co-immunoprecipitations were carried out under basal and ER stress conditions, either from HEK293T control cells or cells treated with tunicamycin (Appendix Fig S3A), or from a mouse ligated colonic loop model before and after treatment with tunicamycin (Fig 3A). The mouse colon was chosen as both AGR2 and TMED2 are highly expressed in this tissue. Both *in vitro* and *in vivo*, AGR2 was found in a complex with TMED2 that dissociated upon ER stress (Fig 3A and Appendix Fig S3A). This observation suggests that under basal

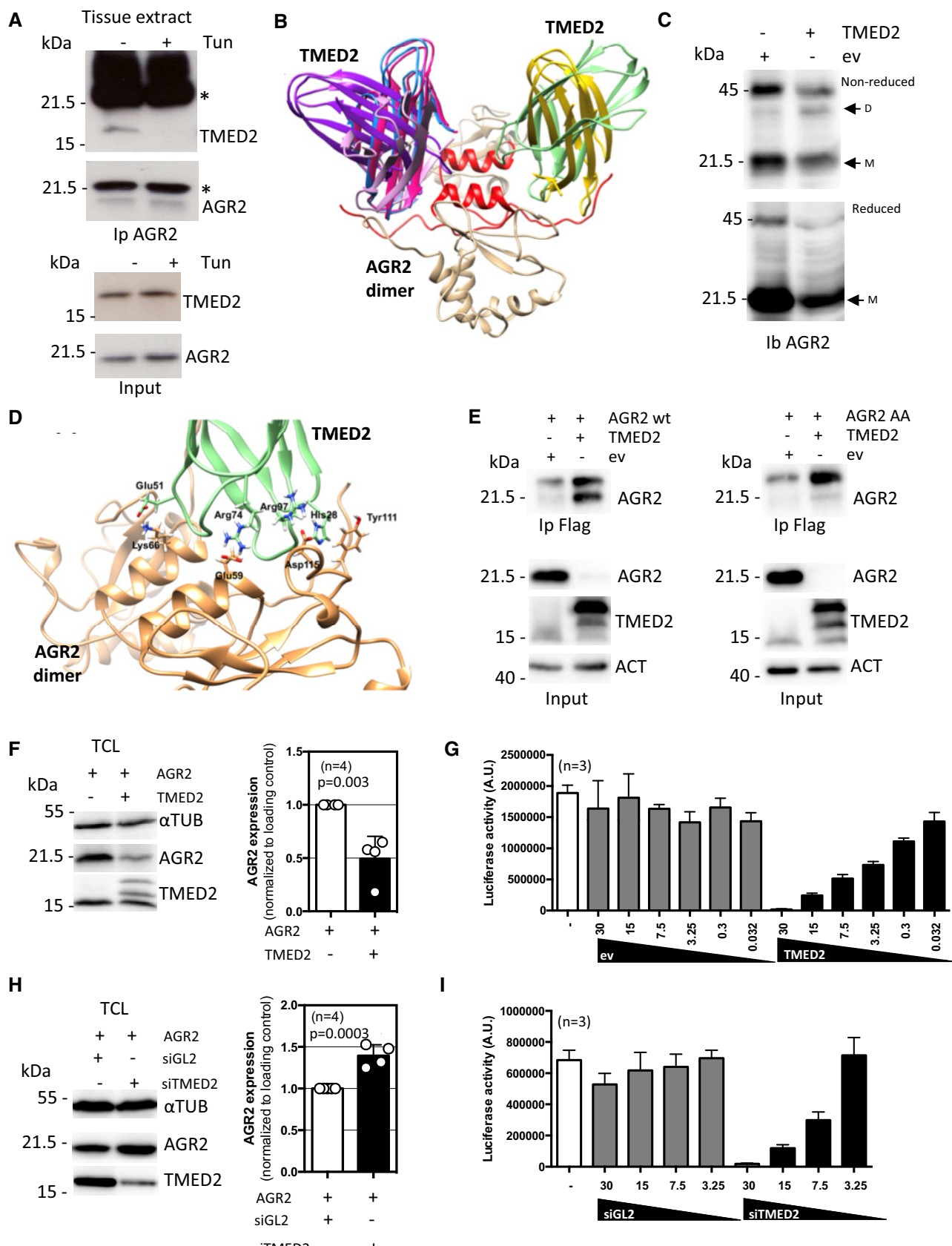

**Figure 3.**

**Figure 3. Validation of AGR2–TMED2 interaction and its impact on AGR2 dimerization./TMED2 functional relationships with AGR2.**

A   Co-immunoprecipitation of AGR2 with TMED2 under basal and tunicamycin induced ER stress in intestinal epithelial cells treated *in situ*. *indicates Immunoglobulin heavy and light chains.

B   Structural association of TMED2 and AGR2. The six TMED2 orientations (pink, dark pink, purple, blue, yellow, and green) best positioned to stabilize the AGR2 dimer (colored tan with N-termini and dimer interface alpha helix in red).

C   Changes in AGR2 dimer and monomer ratio in TMED2 overexpressing cells. DSP-stabilized AGR2 was analyzed under non-reducing (top blot) or reducing conditions (bottom blot). D = dimeric, M = monomeric AGR2.

D   Specific interactions between residues in AGR2 dimer (tan) and best oriented TMED2 structure (green). Glu51, Arg74, and Arg97 of TMED2 form ionic interactions with Lys66, Glu59, and Asp97 of AGR2. His28 of TMED2 interacts with Tyr111 of AGR2 through π-π-interactions.

E   Co-immunoprecipitation of AGR2 wild-type (WT, left panel) or AGR2 AA mutant (right panel) with TMED2 in HEK293T cells. Immunoprecipitation was performed with mouse anti-Flag antibody to pull-down the ectopic protein tag.

F   Western blot analysis (left panel) and quantification (right panel) of AGR2 intracellular expression upon TMED2 overexpression. Data are representative of four independent experiments. Tubulin (TUB) was used as a loading control. The graph represents average signal normalized on reporter protein expression ±SD. The Mann–Whitney statistical test was used.

G   HEK293T cells were transfected with control (ev) or TMED2 plasmid and then used in the ERMIT assay with AGR2 WT as described in Fig 1E ($n$ = 3). The graph represents average signal normalized on reporter protein expression ±SD.

H   Western blot analysis (left panel) and quantification (right panel) of AGR2 intracellular expression in total cell lysate (TCL) upon TMED2 silencing. Data are representative of 4 independent experiments. Tubulin (TUB) was used as a loading control. The graph represents average signal normalized on reporter protein expression ±SD. The Mann–Whitney statistical test was used.

I   HEK293T cells were silencing for TMED2 and then used in the ERMIT assay with AGR2 WT as described in Fig 1E ($n$ = 3). The graph represents average signal normalized on reporter protein expression ±SD.

and stress conditions AGR2 is present in different functional complexes, a result supported by our siRNA and proteomic screens (Higa *et al*, 2011), where AGR2 mainly contributed to import into the ER, export to the Golgi apparatus or to ERAD (Appendix Fig S2D). The possible interaction of AGR2 monomer and dimer with TMED2 was explored using extensive protein–protein docking (Fig 3B and Appendix Fig S3B–E). The identified interaction orientations between TMED2 and AGR2 monomer are for the most part unstable and will block the possibility of AGR2 dimer formation (Appendix Fig S3B). Docking between TMED2 and AGR2 dimer, on the other hand, rendered several conformers in which TMED2 simultaneously interacts with both AGR2 monomers in the N-terminal/dimer interface regions (Fig 3B), and where perfect complementarity between structures and electrostatic surfaces of the two are noted (Appendix Fig S3C–E). We next examined the mechanisms underlying TMED2 regulation of AGR2 homodimerization. TMED2 overexpression led to enhanced AGR2 homodimer formation as evaluated using DSP-mediated cross-linking (Fig 3C). To further characterize the functional role of the interaction between TMED2 and AGR2, we sought to generate a mutant AGR2 unable to interact with TMED2, thereby not directly affecting TMED2 functions. To this end, a molecular modeling approach was undertaken to identify amino acid residues involved in the TMED2-AGR2 interaction and revealed that K66 and Y111 might play key roles (Fig 3D). As such, K66 and Y111 were mutated to alanine residues (referred to as AGR2 AA hereafter) and the interaction between AGR2 and TMED2 was evaluated using co-immunoprecipitation. As expected, whereas AGR2wt and TMED2 co-immunoprecipitated, the interaction between TMED2 and AGR2 AA was impaired (Fig 3E). Of note, the expression levels of AGR2 AA were lower than those of AGR2 WT, suggesting that TMED2 might also regulate the expression of this protein independently of their interaction. We next monitored the impact of TMED2 expression alteration on AGR2 level. Interestingly, overexpression of TMED2 led to reduced expression of AGR2 (Fig 3F), and reduced ERMIT signals, correlative to the loss of expression (Fig 3G). In contrast, the silencing of TMED2 led to enhanced expression of AGR2 (Fig 3H), but decreased ERMIT signals, indicative of effective dimerization inhibition (Fig 3I).

## AGR2 dimerization ability does not affect its chaperone functions but alters its localization

To explore the functional relevance of AGR2 dimerization, we tested how AGR2 regulates cargo secretion. As such the previously described interactions of AGR2 with the two plasma-membrane GPI-anchored proteins CD59 and LYPD3, that were reported in proteomics studies, were confirmed using ERMIT with the monomeric AGR2 E60A used as bait and either CD59 or LYPD3 used as preys, OS9 was used as a negative control (Fig 4A). Furthermore, we monitored the AGR2 contribution to the ER quality control and protein secretion using CD59 WT and mutant form, CD59 C94S. The latter due to its misfolding is no longer efficiently exported to the cell membrane and accumulates in the ER lumen (Fig 4B) even though the expression levels are similar (Fig 4C). We also found that both AGR2 WT and AA interacted with GFP-CD59 WT or C94S (Fig 4D and E). Interestingly, the modification of AGR2 and TMED2 expression levels impacted on CD59 degradation and trafficking (Fig 4F and G). Indeed, although AGR2 silencing led to reduced expression of intracellular CD59 WT (25%) and CD59 C94S (50%), it did not impact further on the expression of both proteins at the cell surface, thereby suggesting a role of intracellular AGR2 in quality control in the ER (Fig 4F and G). TMED2 silencing led to reduced expression of intracellular CD59 (either WT or C94S), and a similar effect was observed for cell surface expression (Fig 4F and G, and Appendix Fig S4A and B). These data indicated that the interplay between AGR2 and TMED2 exerts a selective regulation on protein folding and trafficking and contributes to protein quality control in the ER. To test the functionality of AGR2 AA, rescue experiments were carried out and showed that overexpression of either AGR2 WT or AGR2 AA restored the expression of GFP-CD59 (WT or C94S) total and at the cell surface (Fig 4H and I), thereby indicating that AGR2 AA conserved its ability to participate to ER folding and quality control mechanisms.

Further, we sought to investigate the impact of AGR2 on the secretion of cargo proteins under normal and ER stress conditions. Given that AGR2 peptide binding sites are present on alpha-1-antitrypsin (A1AT; Appendix Fig S4C), we tested if AGR2 impacts on the secretion of this cargo. We examined the effect of silencing of

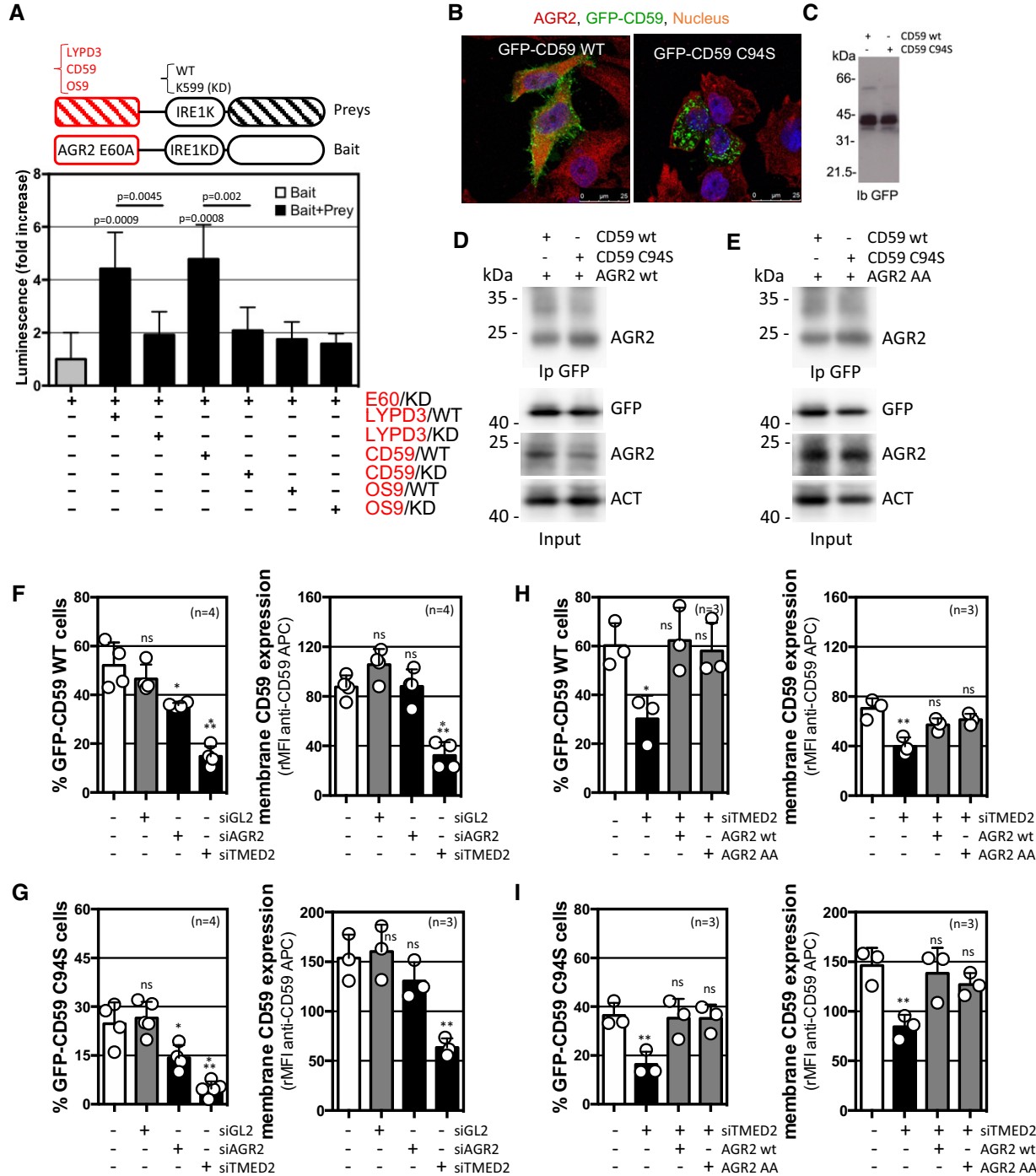

**Figure 4. AGR2 status alteration impacts on client protein biogenesis.**

A   The ERMIT-based validation of the complex formation between AGR2 and LYPD3 or CD59, but not OS9. The monomeric AGR2 E60A was used as bait, while LYPD3, CD59, and OS9 were used as prays (*n* = 6). The graph represents average signal normalized on reporter protein expression ±SD. For statistical analyses, the Mann–Whitney test was used.

B, C   Expression and subcellular localization of CD59-GFP WT and C94S.

D, E   Co-immunoprecipitation of CD59-GFP WT or CD59 C94S mutant with AGR2 WT (D) or AGR2 AA mutant (E) in HEK293T cells.

F–I   FACS analysis of CD59 WT (F and G) or CD59 C94S (H and I) total (F and H) and membrane (G and I) expression in HEK293T cells silenced for AGR2 (siAGR2) and TMED2 (siTMED2) when compared to control (siGL2) cells (F and H) or silenced for TMED2 (siTMED2) and overexpressing AGR2 WT or AGR2 AA mutant (G and I). Data are representative of three to four independent experiments. Membranous CD59 expression was detected with anti-GFP APC. ns: non-statistically significant; for (F) (*): *P* = 0.0385, (***): *P* = 0.0003 (left panel) and *P* = 0.0002 (right panel), respectively; for (G) (*): *P* = 0.0334, (**) *P* = 0.0036, and (***) *P* = 0.0003; for (H) (*): *P* = 0.0171 and (**): *P* = 0.0084; and for (I) (**) *P* = 0.0086 (left panel) and *P* = 0.0076 (right panel), respectively. The graph represents average GFP signal ±SD. For statistical analyses, the Mann–Whitney test was used.

AGR2 on secretion of A1AT by immunoblot under basal and stress conditions (Appendix Fig S4D and E). Under basal conditions, AGR2 was not involved in the secretion of A1AT as the silencing of AGR2 did not affect the kinetics of A1AT secretion. However, upon ER stress the retention of A1AT in the ER was decreased in the absence of AGR2. This suggests that AGR2 might also be involved in sensing ER homeostasis. Lastly, as shown in Appendix Fig S4F, the presence of AGR2 stabilized the expression of MUC2 in HT29 cells, further confirming a crucial role for AGR2 in ER proteostasis. In addition, treatment of HT29 cells with the PTTIYY peptide (AGR2 binding; Clarke *et al*, 2011) rescued MUC2 expression upon ER stress (Appendix Fig S4G), suggesting the importance of the AGR2/MUC2 interaction in MUC2 quality control.

Since we observed an impact of TMED2 expression changes on iAGR2 expression levels, we sought to investigate the underlying molecular mechanisms involved in this phenomenon. First, the effects of overexpression of TMED2, which seemed to decrease the levels of intracellular AGR2 (iAGR2; Fig 3), were not reversed by ERAD pharmacological inhibitors (Appendix Fig S5A). However, we found that this occurred through an alternative degradation mechanism involving autophagy (Fig 5A and B) and was reversed by chloroquine treatments (Fig 5C and D). This pointed toward an lysosomal/autophagy-dependent degradation of AGR2 induced by TMED2 overexpression. However, when we tested the presence of AGR2 in the extracellular milieu, we detected an anti-AGR2 immuno-reactive band with an unexpected electrophoretic mobility (~37 kDa; Appendix Fig S5E). This indicated that cells overexpressing TMED2 might present aberrant secretion features. This was confirmed by analyzing the insoluble material released by TMED2 overexpressing cells using cryo-electron microscopy that presented a very heterogenous profile of extracellular vesicles (and CD63 staining) compared to control cells (Fig 5F). Collectively, these data show that overexpression of TMED2 leads to the abnormal secretory features including the release of aberrant AGR2 entities. TMED2 silencing, on the other hand, resulted in the increase of the intracellular fraction of AGR2 (iAGR2, Fig 3) and promoted elevated AGR2

secretion in the medium (eAGR2; Fig 5G and H). Finally, we tested how constitutively monomeric (E60A) or dimeric (Δ45) AGR2 form behaved regarding secretion. Our results indicate that AGR2 E60A was secreted more efficiently than AGR2 WT and in the contrary, AGR2 Δ45 was retained inside the cell (Fig 5I). Importantly, TMED2 overexpression or silencing did not impact further the secretion of AGR2 AA (Appendix Fig S5B and C) thereby demonstrating the dependency of AGR2/TMED2 interactions for AGR2 secretion. Together, these results indicate that alteration of AGR2 dimeric versus monomeric status impacts on AGR2 release in the extracellular milieu (either as a part of an altered secretory material or as a monomer). At last, TMED2 overexpression led to the release of AGR2 in a way that suggested the involvement of the endo-lysosomal system. This was also observed upon treatment of the cells with Brefeldin A that prompted a bafilomycin A1-sensitive release of AGR2 (Appendix Fig S5D). The alternative release of AGR2 might therefore occur through a non-conventional secretion mechanism.

## Pathophysiological implication of AGR2 dimerization control

Since AGR2 was involved in hypersensitivity of intestinal epithelium to inflammation (Zhao *et al*, 2010) and since TMED2 was found to regulate AGR2 dimeric status, we postulated that mice exhibiting altered TMED2 expression should also display an intestinal phenotype. To test this hypothesis, we evaluated the expression of AGR2 and MUC2 in the intestine of mice expressing lower levels of TMED2 (heterozygous deficient; Hou *et al*, 2017). Interestingly, typical signs of chronic intestinal inflammation were observed in TMED2 hypomorph mice such as loss of mucosecretion, inflammatory cell infiltrate, and hyperproliferation of mucosa in both the proximal colon and ileum (Fig 6A). These observations were associated with apparent lymphocytosis (Fig 6A). Furthermore, TMED2 hypomorph mice exhibited lower global expression level of both AGR2 and MUC2 than WT mice (Fig 6B and C), thereby partly phenocopying the results observed in AGR2 deficient mice. As we recently showed that eAGR2 could exert signaling properties on cells by inducing

---

**Figure 5. Cellular mechanisms of AGR2 secretion.**

A  Formation of GFP-LC3 autophagic puncta in TMED2 overexpressing HEK239T cells as monitored using confocal microscopy (right panel). DAPI was used for nuclear staining visualization. Percentage of GFP-LC3 puncta in control (CTL) and TMED2 overexpressing HEK293T cells as quantified from three independent experiments by counting 240 GFP-LC3 positive cells for each condition (left panel). (**): $P = 0.0032$. The graph represents average western blot normalized signal ±SD. For statistical analyses, the Mann–Whitney test was used.

B  Western blot detection (left panel) and quantification (right panel) of LC3 level in control and TMED2 overexpressing HEK293T cells treated or not with 50 μM chloroquine for 2 h. Actin (ACT) served as a loading control. (*): $P = 0.0498$. The graph represents average western blot normalized signal ±SD. For statistical analyses, the Mann–Whitney test was used. $n = 4$.

C  Western blot analysis of AGR2 expression in control and TMED2 overexpressing HEK293T cells upon autophagy inhibition with 20 and 10 μM chloroquine treatment. Actin (ACT) served as a loading control.

D  HEK293T cells were transfected with control (ev) or TMED2 plasmid and then used in the ERMIT assay with AGR2 WT as described in Fig 1E in the presence of gradual amounts of chloroquine. The graph represents average signal ±SD. For statistical analyses, the Mann–Whitney test was used, $n = 4$.

E  Western blot analysis of secreted AGR2 upon TMED2 overexpression. eAGR2, extracellular AGR2; iAGE2, intracellular AGR2.

F  Representative pictures of extracellular vesicles (EVs) heterogeneity purified from conditioned media of HEK293T cells transfected with control (ev) or TMED2 plasmid and analyzed by cryo-electron microscopy. Right upper panel: Western blot analysis of CD63 in total medium (left) and in the purified extracellular vesicles fraction purified from media conditioned by control and TMED2 overexpressing cells (middle) and Western blot analysis of AGR2 levels (right) in extracellular vesicles enriched from culture media conditioned by control and TMED2 overexpressing cells.

G  Western blot analysis of secreted AGR2 upon TMED2 silencing eAGR2, extracellular AGR2; iAGE2, intracellular AGR2.

H  Quantification of extracellular AGR2 level (eAGR2) in cells silenced for TMED2 (siTMED2), compared to control (siGL2). Data are representative of three independent experiments and are presented as extracellular-to-intracellular AGR2 ratio. (**): $P = 0.0029$. The graph represents average western blot normalized signal ±SD. For statistical analyses, the Mann–Whitney test was used.

I  Secretion of AGR2 wild-type (wt), E60A, and Δ45 mutant as determined by Western blot. eAGR2, extracellular AGR2; iAGE2, intracellular AGR2.

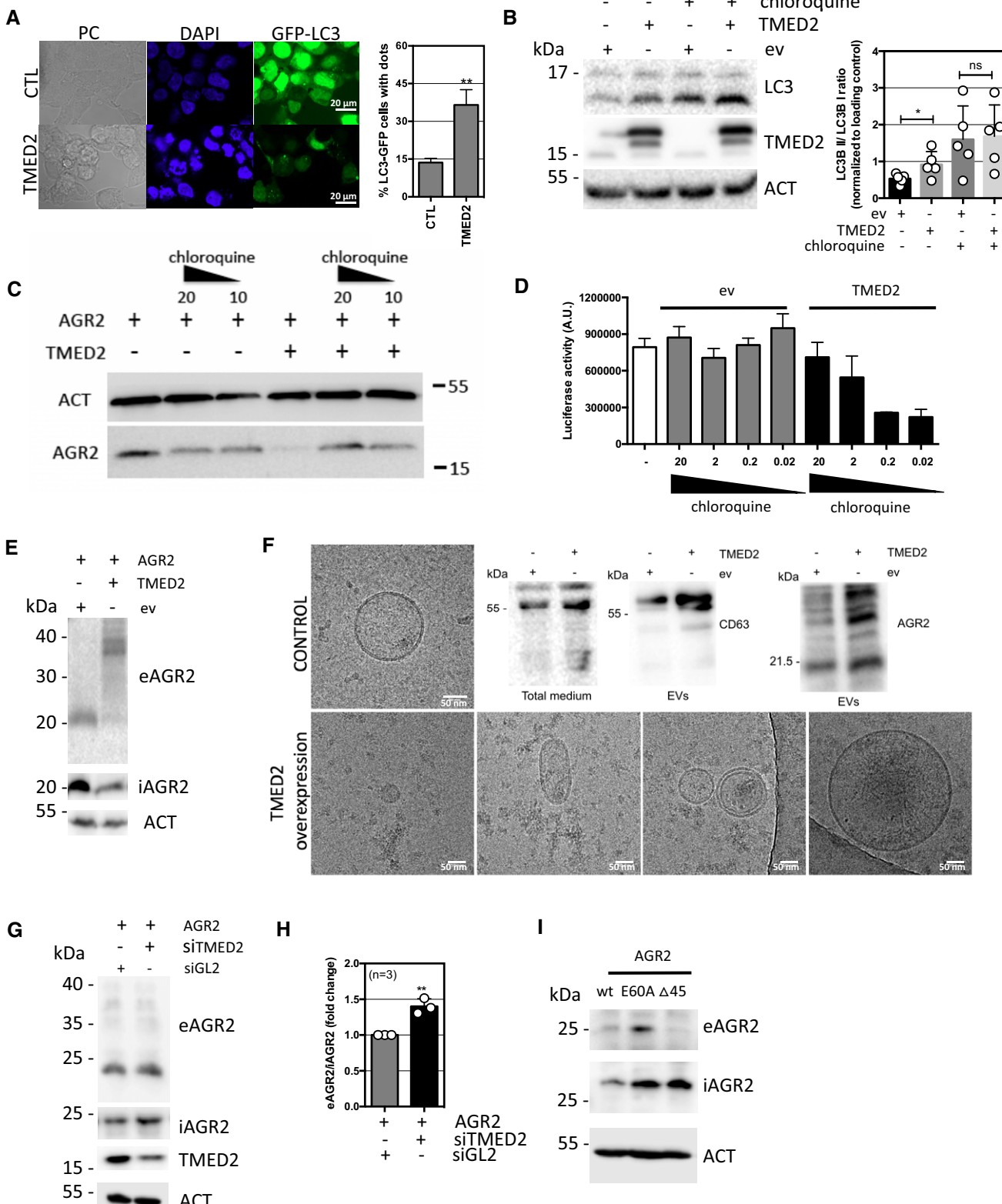

**Figure 5.**

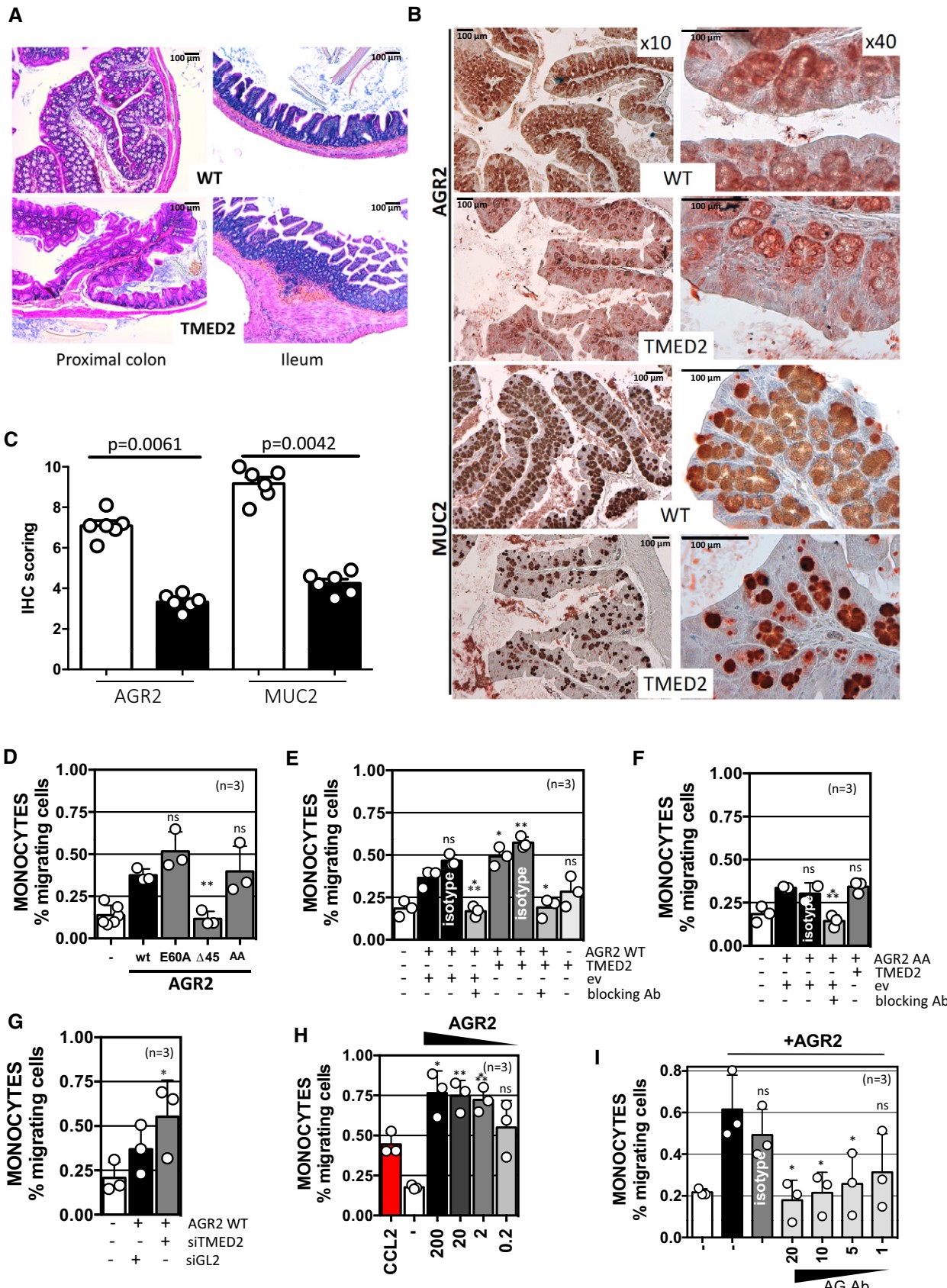

**Figure 6.**

◄

**Figure 6.  Alteration of TMED2 expression and relationships with eAGR2-mediated monocytes attraction.**

A    Representative histological H&E staining of sections of the proximal colon and ileum of WT and TMED2 hypomorph mice.

B    Immunohistological analysis of AGR2 and MUC2 expression in the proximal colon of WT and TMED2 hypomorph mice.

C    Semi-quantitative analysis of AGR2 and MUC2 expression in the proximal colon of WT (blank bars) and TMED2 hypomorph mice (black bars) (*n* = 6). The graph represents average IHC signal ±SD. For statistical analyses, the Mann–Whitney test was used.

D    Freshly isolated PBMCs were placed in Boyden chambers toward media conditioned by cells overexpressing AGR2 WT, E60A, Δ45, or AGR2 AA mutants and incubated for 24 h. Migrating cells were then characterized and quantified by flow cytometry using FCS/SSC parameters or CD14 monocyte marker. (**): *P* = 0.0016. The graph represents average IHC signal ±SD. For statistical analyses, the Mann–Whitney test was used.

E, F  Freshly isolated PBMCs were placed in Boyden chambers toward media conditioned by cells overexpressing TMED2 and either AGR2 WT (E) or AGR2 AA mutant (F) and incubated for 24 h. Migrating cells were then characterized and quantified as indicated in Fig 6D. For (E) (*): *P* = 0.0461 (AGR2WT/TMED2) and 0.018 (AGR2WT/TMED2 + blocking Ab) respectively; (**): *P* = 0.0053 and (***): *P* = 0.0048; for (F) (***): *P* = 0.0007. The graph represents average IHC signal ±SD. For statistical analyses, the Mann–Whitney test was used.

G    Freshly isolated PBMCs were placed in Boyden chambers toward media conditioned by cells silenced for TMED2 and incubated for 24 h. Migrating cells were then characterized and quantified as indicated in Fig 6D. (*): *P* = 0.0149. The graph represents average IHC signal ±SD. For statistical analyses, the Mann–Whitney test was used.

H    Freshly isolated PBMCs were placed in Boyden chambers toward decreased doses of recombinant AGR2 and incubated for 24 h. CCL2 cytokine was used as positive control for monocyte migration. ns: non-statistically significant, (*): *P* = 0.0171, (**): *P* = 0.0084, (***): *P* = 0.0003. The graph represents average IHC signal ±SD. For statistical analyses, the Mann–Whitney test was used.

I    Impact of AGR2 blocking antibodies on AGR2-mediated monocytes migration was tested using Boyden chambers as described above. The concentrations of recombinant human AGR2 were of 200 ng/ml, and decreasing amounts of antibodies were used from 20 μg to 1 μg. The non-relevant antibody (Isotype) was used at the maximal dose of 20 μg. Data are representative of three independent experiments. ns: non-statistically significant, (*): *P* = 0.0165 (Ab = 20 μg/ml), 0.0223 (Ab = 10 μg/ml) and 0.0493 (Ab = 5 μg/ml), respectively. The graph represents average IHC signal ±SD. For statistical analyses, the Mann–Whitney test was used.

EMT programs (Fessart *et al*, 2016), and since in our cellular models TMED2 silencing led to enhanced released of eAGR2, we reasoned that eAGR2 might also play a role in the chemoattraction of pro-inflammatory cells. To determine the direct involvement of eAGR2 in chemoattraction, PBMCs purified from three independent healthy donors were exposed either to media conditioned by cells overexpressing AGR2 WT, E60A, Δ45, or AA. Chemoattraction of monocytes from PBMCs was observed only when AGR2 was found in the extracellular milieu, namely when conditioned media from cells transfected with AGR2 WT, E60A, or AA was used (Fig 6D). Similar results were obtained when using media from cells overexpressing AGR2 WT or AA and simultaneously overexpressing TMED2 (Fig 6E and F), media from cells silenced for TMED2 (Fig 6G) or even recombinant human AGR2 (Fig 6H). Remarkably, AGR2 blocking antibodies were able in all cases to impede monocytes migration (Fig 6E, F and I). These experiments revealed that in all cases, eAGR2 was able to selectively promote monocyte attraction, thereby linking eAGR2 to pro-inflammatory phenotypes and unraveling the extracellular gain-of-function of AGR2 as a pro-inflammatory chemokine. Collectively, our results link the interaction between TMED2 and AGR2 and by extend the monomeric versus dimeric status of AGR2 to pro-inflammatory phenotypes in the intestine. To test the relevance of these results in human IBD, we first evaluated the expression levels of the pathophysiological relevance of AGR2 dimer regulators in colonic biopsies from patients with IBD. Fifty-two of the 71 candidates presented in Appendix Table S1 were first tested in non-inflamed colonic biopsies from healthy controls, patients with ulcerative colitis (UC) and patients with Crohn's disease (CD), the two main classes of IBDs (Appendix Table S3). Messenger RNA expression levels of 12 out of 52 genes were found to be significantly different in CD while only 3 showed significant differences in UC (Appendix Fig S6). The expression differences in AGR2 modulators were exacerbated in colonic CD patients (CC; Fig 7A). To corroborate these findings, a validation cohort consisting of healthy controls and patients with ileo-colonic CD was used to evaluate mRNA expression levels of the 52 genes of interest. Fourteen genes, including the 12 genes previously

identified, were significantly different in patients with CD, supporting the initial findings (Appendix Fig S6B). This allowed for the differentiation of CD patients from healthy controls (Fig 7B). Moreover, a functional enrichment analysis revealed that 6 genes whose silencing disrupted AGR2 dimer formation were either up-regulated or down-regulated in CD (namely TMED2, RPN1, KTN1, LMAN1, AMFR, AKAP6) and that 4 genes whose silencing promoted AGR2 dimerization were systematically down-regulated in CD (namely P4HTM, SYVN3, CES3, SCAP). TMED2 mRNA (Fig 7C) and protein (Fig 7D) expression was increased in CD, mainly in normal intestinal epithelial cells. TMED2 overexpression was detected in patients with active (A) CD and correlated with high recruitment of CD163 positive macrophages in the colonic mucosa (Fig 6E and F). Remarkably, patients with quiescent (Q) CD exhibited a moderate loss of AGR2 global staining which likely correlated with its probable secretion (Fig 7E and F). These data indicate that regulation of AGR2 dimerization is associated with pro-inflammatory responses and enrichment of macrophages in the colonic mucosa that could be observed in CD (Fig 7G). Dissecting the diversity and the local distribution of functional macrophages in patients with active or quiescent CD will further define clinical relevance of AGR2.

## Discussion

The results presented in this study show that in the ER, AGR2 exists under monomeric or dimeric configurations and modulation of AGR2 dimeric versus monomeric status might represent a novel ER proteostasis sensor mechanism in intestinal epithelial cells. Moreover, we identify a mechanism of regulation of AGR2 dimerization through an interaction with the protein TMED2. Furthermore, our data link the perturbation of AGR2 dimerization to inflammatory bowel disease in human in part through the unexpected intervention of AGR2 in the recruitment of inflammatory cells. Collectively, our results document a molecular link between ER proteostasis control and a pro-inflammatory systemic stress response which when abnormal turns out as a disease state in the colon.

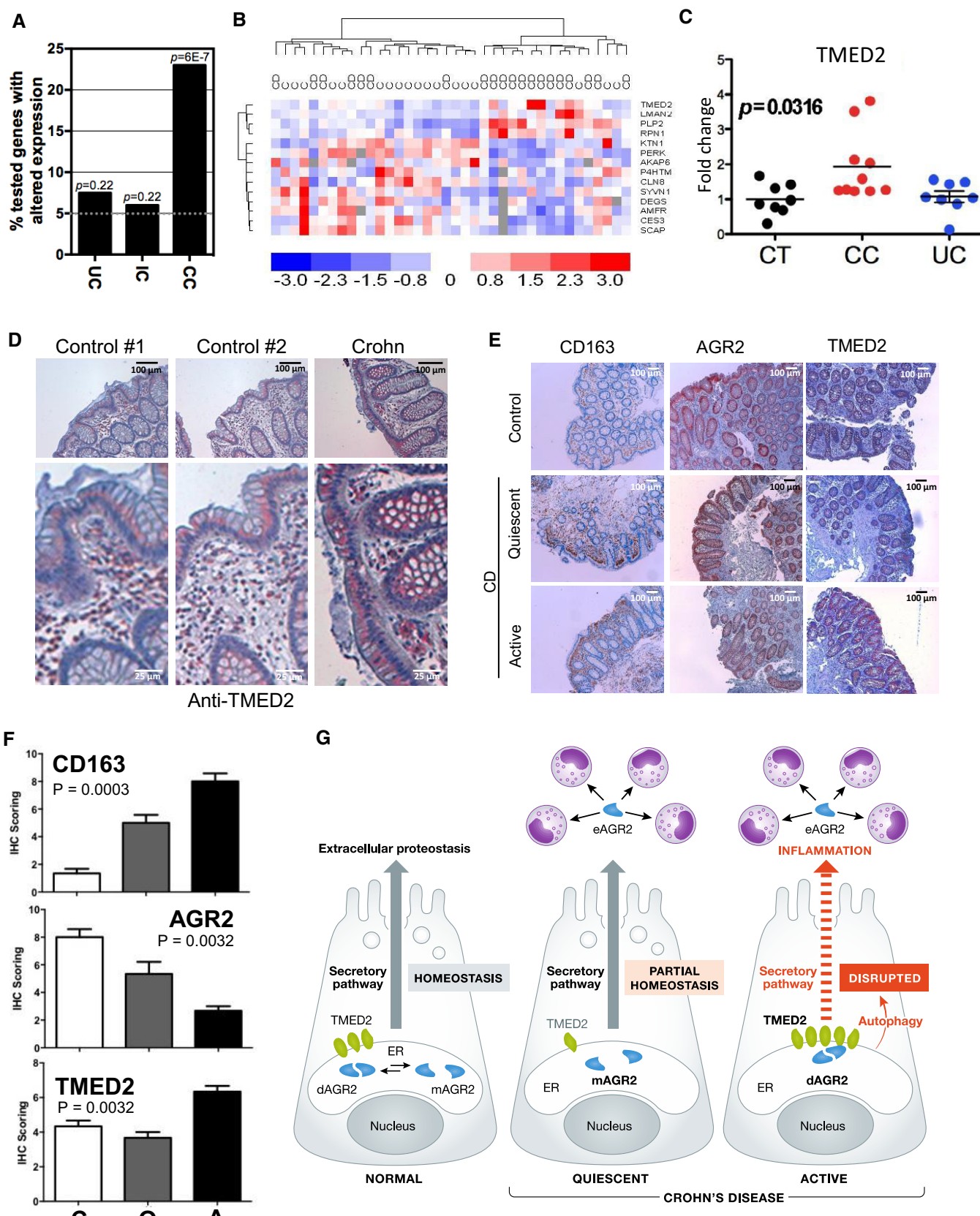

Figure 7.

◀

**Figure 7. Expression levels of AGR2, TMED2, and CD163 in biopsies from healthy controls and patients with IBD.**

A   Selective enrichment of AGR2 modulators in colonic Crohn's disease (CD). Percentage of tested genes with altered expression in colonic biopsies from patients with ulcerative colitis (UC, $n = 8$) and colonic CD (CC, $n = 15$) and in ileal biopsies from patients with ileo-colonic CD (IC, $n = 9$). $t$-test was used.

B   Hierarchical clustering of the AGR2 modulator genes significantly deregulated in non-inflamed colonic mucosa from patients with Crohn's Disease (CD) versus healthy controls (C) (dChip software $t$-test). Gene expression (blue: fold decrease; red: fold increase).

C   TMED2 mRNA expression in healthy controls (CT), colonic Crohn's disease (CC), and ulcerative colitis (UC). ANOVA was used.

D   Representative immunohistological analysis of TMED2 in non-inflamed colonic biopsies from healthy controls and patients with active CD. Scale bars: 100 μm.

E   Representative immunohistological analysis of CD163 (scavenger receptor present in macrophages), AGR2, and TMED2 in non-inflamed colonic biopsies from healthy controls and patients with active or quiescent CD.

F   Semi-quantitative analysis of CD163, AGR2, and TMED2 immunostaining in colonic mucosa from healthy controls (C) and patients with quiescent (Q) or active (A) CD. Three random sections from each patient of the four groups were scored. Final IHC scores (A × B range to 0–9) combine the percentage of positively labeled cells. A: % of IHC positive labeled cells—0 (0%), 1 (< 30%), 2 (30–60%), 3 (> 60%), and the intensity of the reaction product. B: 0 (no reaction), 1 (weak), 2 (mild), 3 (strong) in most of the examined fields. All values are mean ± SEM. $n = 12$, ANOVA was used as a statistical test.

G   Schematic representation of the role of AGR2 dimer alteration in intestinal epithelial cells under basal conditions or in Crohn's disease. In normal cells, the ratio of dimeric versus monomeric AGR2 defines proteostasis boundaries in the ER, and those are altered in diseased conditions through the alteration of the expression of AGR2 dimer regulators, thereby leading to pro-inflammatory signals comprising the release of AGR2 in the extracellular milieu and to the induction of uncontrolled autophagy and the subsequent alteration of protein secretion.

We first reasoned that since an excess of AGR2 dimers or AGR2 monomers yields a pro-inflammatory response, a systemic adaptive reaction, the relative concentrations of each form might be linked to proper function of the early secretory pathway. In this context, dysregulation of the relative equilibrium of AGR2 dimers and monomers could be a sign of ER proteostasis imbalance. In the context of IBD, protein homeostasis within the early secretory pathway and its adaptation to the perturbation through the UPR have been shown to play instrumental roles in disease onset (Grootjans *et al*, 2016). In the present work, we identified AGR2 as a critical player in such adaptive mechanism and we further demonstrated that under basal conditions AGR2 mainly interacts with Golgi export components to ensure proper protein folding, while during ER stress it forms functional complexes with ERAD machinery to clear the misfolded proteins from the ER. Moreover, this study provides the identification of AGR2 status, monomer versus dimer balance, as an early event possibly able to define the extent and some characteristics of intestinal inflammation. This is particularly appealing for IBD, which is characterized by the chronic inflammation and ulceration of the gastrointestinal tract due to an overactive immune digestive system. Our data suggest that perturbation of AGR2 dimerization, due to variable expression levels of its client proteins, can lead to IBD development. This could actually be relevant at several levels through the release of extracellular AGR2 which might as previously found in other models induce Epithelial-to-Mesenchymal Transition markers[13] to promote fibrosis which is a hallmark of Crohn's disease and in the mean-time to promote the recruitment of macrophages to the site of damage to precondition the tissue for uncontrolled inflammation.

Interestingly, our results also establish that the interaction between AGR2 and TMED2 plays a key role in AGR2 dimerization control by stabilizing the dimer. The alteration of TMED2 expression in mice, resulting from the heterozygous expression of a mutant form of the protein that is not properly synthesized (Hou *et al*, 2017), resulted in alteration of colon homeostasis and inflammation. Moreover, overexpression of TMED2 was detected in active CD and may also be associated with inflammation through autophagy-dependent AGR2 release in the extracellular milieu (Park *et al*, 2009; Zhao *et al*, 2010). A similar mechanism could be applied to other AGR2 expressing cells, such as pancreatic, biliary, or lung epithelia. Findings from this study might be further applicable to

cancer biology, since proteostasis imbalance has emerged as a major cancer hallmark, capable of driving tumor aggressiveness (Chevet *et al*, 2015). In light of our findings, control of AGR2 dimerization may well be a relevant factor in cancer development. High AGR2 expression, as well as its secretion into body fluids, was reported in many cancer types and associated with pro-tumorigenic phenotype and poor patient outcome (Chevet *et al*, 2013; Brychtova *et al*, 2015; Obacz *et al*, 2015). However, questions remain as to what is the predominant form of AGR2 in cancer cells, how is the formation of AGR2 dimer versus monomer precisely regulated, and what are the biological/functional consequences of AGR2 dimerization? These issues warrant deeper investigation. Collectively, our data provide the first evidence of the existence of ER sensors, such as AGR2, that contribute to the regulation of proteostasis boundaries in this compartment, and whose alteration leads to pro-inflammatory responses.

## Materials and Methods

### Materials

Tunicamycin (used at 2 μg/ml or otherwise indicated) was from Calbiochem (Guyancourt, France), thapsigargin (used at 500 nM or otherwise indicated) was from Calbiochem, Azetidine-2-carboxylic acid (used at 5 mM or otherwise indicated), and DTT (used at 0.5 mM or otherwise indicated) was from Sigma (St. Louis, MO, USA). The siRNA library was from RNAi (http://rnai.co.jp/lsci/products.html). DSP was from Thermo Fisher—Pierce (Villebon-sur-Yvette, France).

### Plasmid constructs

Constructs used in this report were derived from the pcDNA5/FRT/TO (Invitrogen) plasmid. The segment encoding the transmembrane and cytosolic domains of IRE1 was cloned in pcDNA5/FRT/TO plasmid by standard PCR and restriction-based cloning procedures. All used primers are listed in Appendix Table S4. Baits and preys present in the hORFeome v8.1 were directly transferred in the pcDNA5/FRT/TO/IRE1 using the Gateway™ cloning technology (Life Technologies). The mutant constructions were obtained by

PCR mutagenesis with the QuickChange® II Site-Directed Mutagenesis Kit (Agilent Technologies) using the different primers presented in Appendix Table S4. The XBP1 splicing reporter was described previously (Samali *et al*, 2010). The hTMED2 expression plasmid was obtained from Sino Biological (HG13834-CF). The GFP-LC3 plasmid was a kind gift from Dr P. Codogno (Paris, France). AGR2 cDNA (WT, E60A, Δ45, and AA) were obtained from Genewiz (Sigma-Aldrich) and were cloned in pcDNA3.1 plasmids.

### SiRNA screening

The screen was performed using a custom-made siRNA library targeting 274 ER-resident proteins. Five thousand HEK293T cells were seeded in black 96-well plates. One day later, the cells were transfected with 200 pg of the AGR2/WT bait, 1.5 pmol of siRNA, and 3 ng of the XBP1 luciferase reporter using the calcium phosphate precipitation procedure. In parallel, a counterscreen was performed by transfecting the siRNAs and the XBP1 luciferase reporter in the absence of the AGR2 WT bait. Two days after transfection, the luciferase activity was measured by chemiluminescence in an EnVision Multilabel Plate Reader (PerkinElmer, Waltham, MA, USA). The raw values were log2 transformed and were normalized to the average signal of the plate. The average negative signal of the plate was subtracted, separately for each replicate, and a quantile normalization was performed. *T*-test and Kruskal–Wallis statistical analyses were performed to select the list of significant candidates.

### Patients and sample analyses

The experiments conformed to the principles set out in the WMA Declaration of Helsinki and the Department of Health and Human Services Belmont Report. Human ascending colon and ileal biopsies were obtained from the IBD Gastroenterology Unit, Beaujon Hospital. The protocol was in agreement with the local Ethics Committee (CPP-Ile de France IV No. 2009/17), and written informed consent was obtained from all the patients before inclusion. The clinical characteristics of IBD patients are shown in Appendix Table S3. Thirty-two healthy controls, eight patients with UC, and 40 patients with CD were selected (consecutively between 2012 and 2015) and included in this study. All patients were diagnosed based on classical clinical features as well as radiological, endoscopic, and histological findings. All biopsies were taken from the non-inflamed area of the right colon or the terminal ileum and analyzed by an expert GI pathologist. Unaffected areas were defined as mucosal regions without any macroscopic/endoscopic and histological signs of inflammation. To preserve tissue transcriptional profiles, biopsy specimens were kept at −80°C until RNA extraction.

### Immunohistochemistry

Paraffin-embedded sections of colon were deparaffinized in xylene, rehydrated, incubated in 3% hydrogen peroxide for endogenous peroxidase removal, and heated for 10 min in sub-boiling 10 mM citrate buffer (pH 6.0) for antigen retrieval. Then, sections were processed using the ImmPRESS reagent kit (Vector Laboratories). Primary antibodies against CD163 (AbCam, ab87099), TMED2 (Santa Cruz Biotechnology, sc-376459), and AGR2 (Novus Biologicals, NBP1-05936) were used.

### Cell culture

HEK293T cells were cultured in a 5% $CO_2$ humidified atmosphere at 37°C and grown in Dulbecco's modified Eagle's medium (Invitrogen) with 10% fetal calf serum (Cambrex Corp.). HeLa, HC116, Caco, and HT-29 cells were grown in DMEM, 4 g/l glucose, and 10% FBS.

### Western blot analysis

Expression of the ERMIT baits and preys was detected using the M2 mouse monoclonal antibody against the FLAG-tag. AGR2 was detected with H0010551-M03 mouse monoclonal antibody (Abnova), or with 12275-1-AP rabbit polyclonal antibody (Proteintech), TMED2 with sc-376459, C-8 mouse monoclonal antibody (Santa Cruz Biotechnology), LC3 with D11 XP rabbit monoclonal antibody (Cell Signaling), CD59 with anti-GFP antibody (Santa Cruz, Tebu-Bio), and CD63 with EXOAB-CD63A-1 rabbit polyclonal antibody (System Biosciences). Expression of actin was detected using the A2066 rabbit monoclonal antibody (Sigma, St Louis, MO, USA), tubulin using mouse monoclonal antibody (Sigma) and that of with mouse anti-p97 antibody (BD Transduction Laboratories). All primary antibodies were diluted 1:1,000 in PBS 0.1% Tween. Fluorescent goat anti-mouse and anti-rabbit antibodies (LI-COR IRDye 800 CW and 600 CW) at the dilution of 1:2,000 for detection with the Odyssey imager (LI-COR) or HRP-conjugated secondary antibodies at the dilution of 1:7,000 in PBS 0.1% Tween for chemiluminescence imaging.

### Chemical cross-linking

HEK293T cells were cultured as described above. Cells were washed with ice-cold PBS, and cross-linking was performed by incubating with 1 mM DSP (final concentration) for 20 min on ice. DSP was then quenched with 50 mM Tris–HCl pH 7.5, 150 mM NaCl for an additional 20 min on ice. After extensive washing with ice-cold PBS, cells were lysed using RIPA buffer and analyzed by Western blotting.

### Secretory protein productive folding and secretion analysis

Cells were pulse-labeled with 100 μCi/60 mm dish of EXPRE35S Protein Labelling Mix (PerkinElmer) for 30 min. Labeled cells were washed twice with ice-cold PBS and chased in complete medium (DMEM containing 10% FBS) for 0–4 h. Pulse-labeling and chase were also carried out in the presence and absence of 5 mM Azetidine-2-carboxylic acid (Azc). Pulse-labeled cells were washed twice with PBS and incubated in lysis buffer (30 mM Tris–HCl pH 7.5, 150 mM NaCl and 1% Triton X-100) for 30 min on ice and centrifuged at 10,000 *g* for 20 min at 4°C. After pre-clearing using protein G Sepharose (GE Healthcare Bio-Sciences), lysates were incubated overnight with anti-AGR2 antibody (1:500) at 4°C. The beads were then added to the immune complexes and precipitated for 40 min at 4°C with gentle rotation and washed five times with lysis buffer. Immunoprecipitates were eluted with Laemmli sample buffer containing 50 mM DTT for 10 min at 70°C. The proteins were analyzed by both X-ray fluorography and immunoblotting and detected using LumiGLO chemiluminescent substrate system (Kirkegaard & Perry Laboratories).

## Autophagy analysis

This was done according to Klionsky *et al* (2016). Briefly, HEK293T cells were transfected with 1 μg C-FLAG tagged expression plasmid coding for TMED2 using Lipofectamine 2000 reagent (Thermo Fisher Scientific) according to the manufacturers' protocol. Twenty-four hours later, cells were transfected with 1 μg expression plasmid coding for GFP-LC3 as described above. Cells were then washed twice with ice-cold PBS and fixed with 4% paraformaldehyde (PFA) for 20 min at room temperature. GFP-positive cells were visualized using confocal microscopy (Leica SP8). For autophagy inhibition, HEK293T cells were co-transfected with 1 μg expression plasmids coding for AGR2 and TMED2 using Lipofectamine 2000 reagent following the manufacturers' protocol. Twenty-four hours post-transfection, chloroquine was added to the cells at a final concentration of 20 and 10 μM and incubated for 24 h. Subsequently, cells were analyzed for AGR2 protein expression by Western blotting with anti-AGR2 antibody. For LC3 protein detection by Western blot, HEK293T cells were transfected with 1 μg expression plasmids coding for TMED2 as described above. Forty-eight hours post-transfections, cells were treated with 50 μM chloroquine for 2 h.

## Monocytes chemoattraction assay

Peripheral blood mononuclear cells (PBMCs) were isolated from healthy donors. PBMCs were washed in RPMI 1% FCS (Life Technologies), placed in Boyden chambers 3 μm ($5 \times 10^5$ cells/chamber in RPMI 1% FCS; Millipore, France) that were placed in RPMI containing increased concentrations of recombinant AGR2 (Raybiotech, Tebu-Bio, Le Perray en Yvelines, France) or conditioned medium from cells overexpressing AGR2 WT, E60A, Δ45, AA mutants or/and TMED2 and silenced or not for TMED2; and then incubated at 37°C for 24 h. The migrated PBMCs (under the Boyden chambers) were collected, washed in PBS, and cells were stained for monocytes, T, B, and NK cell markers (anti-CD14, -CD3, -CD19, and -CD56, respectively) and analyzed by flow cytometry using a FACS-Canto II flow cytometer (BD Biosciences, Le Pont de Claix, France) or an ACEA Novocyte flow cytometer (Ozyme, France). Data were then analyzed using FACSDiva (BD Biosciences) or NovoExpress (Ozyme). The relative number of migrated cells was estimated by flow cytometry by counting the absolute number of cells.

## Animal experiments: ligated-colonic loops assays

Bred in-house 2-month-old male C57BL/6 mice weighing 22–25 g were used. All experiments were approved by and conducted in accordance with the ethical guidelines of the Local Animal Experimentation Committee (Xavier Bichat Medical School) and were in complete compliance with the National Institutes of Health Guide for the Care and Use of Laboratory Animals. Efforts were made to minimize any pain or discomfort, and the minimum number of animals was used. Mice were anesthetized with ketamine/xylazine (100 mg/kg/10 mg/kg) and kept warm with a 37°C warming pad during the assay. Colons were flushed with warm PBS to eliminate stool before surgical ligation. Two contiguous ligated colonic loops per animal (2 cm in length each) were made between the cecum and the rectum with black-braided nylon, leaving the mesenteric blood vessels intact. One hundred microliters of TM solution (1 mg/

ml) or vehicle were injected into the lower and upper ligated colonic loops, respectively, to avoid potential TM solution efflux in the control loop. After incubation for 3 h, the mice were euthanized and the ligated loops were excised from the colon, rinsed with ice-cold PBS, and stored at −80°C before protein extraction. *TMED2 mutant mice*—For TMED2 mutant mice, all procedures and experiments were performed according to the guidelines of the Canadian Council on Animal Care and approved by the Animal Care Committee of the Montreal Children's Hospital. The 99J (TMED2 mutant) mouse line was generated on a C57/BL6J genetic background and maintained on a mixed C3H genetic background (C3HeB/FeJ and C3HeB/FeV) as previously described (Hou *et al*, 2017). Six-month-old male TMED2 mutants ($n = 6$) and their wild-type littermate ($n = 6$) were analyzed.

## Statistical analyses

Graphs and statistical analyses were done using GraphPad Prism 5.0 software (La Jolla, California, USA). When experiment contained two unmatched groups of values, the nonparametric Mann–Whitney test was used for the comparison of means. When experiment contained three groups of values or more, regular one-way analysis of variance (ANOVA) was used for the comparison of multiple means. For each experiment, the ANOVA *P* value is as indicated in brackets. The ANOVA test was followed by a Bonferroni's multiple-comparison post-test, and selected pairs of data were compared. The Bonferroni *P*-value is indicated at the bottom of each figure legend. The nonparametric Spearman test was used to compare quantitative values of expression. Means were considered significantly different if the $P < 0.05$. Significant variations were represented by asterisks above the corresponding bar when comparing the test with the control condition or above the line when comparing the two indicated conditions.

## Functional pathway analysis

The functional analysis and the ranking of genes according to their pivotal role in the underlying molecular network were performed using the BioInfoMiner web application (https://bioinfominer.com, Lhomond *et al*, 2018). The application performs enrichment analysis using the Hypergeometric test to assess the over-representation of Gene Ontology terms annotated to the initial genes of interest. For the correction of *P*-values, the application avoids multiple testing correction methods (Bonferroni, FDR) that tend to promote enrichments with low biological content (i.e., enrichments corresponding to very specific terms, which are annotated by very few genes in the background distribution). Instead, it employs a nonparametric, resampling-based, empirical alternative to multiple testing corrections, which provides a corrected measure for the statistical significance of the enrichments based on their frequencies of observation. Hence, the algorithm prioritizes terms with less frequent enrichments, which tend to represent broader, more systemic functions, represented by larger groups of genes (Chatziioannou & Moulos, 2011; Pilalis & Chatziioannou, 2013; Pilalis *et al*, 2015). The ranking of genes is performed by a graph-theoretical method, which corrects the annotation bias of the Gene Ontology hierarchical network and ranks the related genes according to their regulatory impact in the corrected semantic network (Moutselos

**The paper explained**

**Problem**
The molecular mechanisms underlying inflammatory bowel diseases remain currently not completely understood, and therapeutic options are scarce.

**Results**
Herein, we have identified that a normally endoplasmic reticulum resident protein, member of the protein disulfide isomerase family, AGR2, exhibits tightly regulated secretion mechanisms that are perturbed in Crohn's disease. We demonstrate that the extracellular release of AGR2 leads to the chemoattraction of monocytes, thereby suggesting extracellular AGR2's pro-inflammatory functions, that can be blocked using specific antibodies.

**Impact**
As a consequence, this discovery could represent an appealing therapeutic approach to attenuate the inflammatory burden in Crohn's disease.

*et al*, 2011; Koutsandreas *et al*, 2016). The highly ranked genes are associated with many distinct, cross-talking systemic processes.

**Molecular dynamic simulation of AGR2 WT and E60A mutant**

The E60A mutant (glutamate to alanine) was obtained through *in silico* mutation in Yasara (Krieger & Vriend, 2014). The MD simulations of the AGR2 WT and E60A mutant were performed utilizing the Gromacs 2016.1 package (Hess *et al*, 2008) and the Amber ff03 force field (Wang *et al*, 2000; Duan *et al*, 2003). For each run, the protein was solvated in a cubic box with TIP3P water molecules and a buffer distance of 10 Å to the walls. The system was neutralized by addition of two $Na^+$-ions, and then energy minimized by steepest decent until the force was < 1,000 kJ/mol/nm, with a maximum number of steps of 5,000 to relax bad contacts. By employing periodic boundary conditions, the system was subjected to a position restraint equilibration run, composed of 500 ps in the NVT ensemble followed by 500 ps in the NPT ensemble to relax the water and ions around the protein at constant temperature (300 K) and pressure (1 bar). The leap-frog algorithm with a time step of 2 fs was used for integration of Newton's second law of motion. Constraints on all bonds were applied using the LINCS (Hess *et al*, 1997) algorithm (LINC-iter = 1 and LINC-order = 4). The temperature was controlled by the Nose–Hoover thermostat (Hoover, 1985) with a coupling constant of 0.1 ps, and the Parrinello–Rahman barostat (Parrinello & Rahman, 1981) coupling algorithm was applied to maintain a constant pressure with a coupling time of 2.0 ps and a compressibility of $4.5 \times 10^{-5}$/bar. Electrostatic interactions were treated with the Particle Mesh Ewald (PME; Darden *et al*, 1993; Essmann, 1995) summation method (Fourier spacing = 16 Å and PME-order = 4), with a short-range electrostatic cut-off of 10 Å. The short-range cut-off for van-der-Waals interactions was also set to 10 Å. The neighbor list for non-bonded interactions was updated every 10[th] step through the Verlet cut-off scheme. Following the equilibration run, a 200 ns unrestraint production run in the NPT ensemble was performed, utilizing the same settings described above. All MD simulations were run on the supercomputer cluster

Beskow, provided by the supercomputing center PDC at KTH, Stockholm. All analyses of the provided trajectories were performed with the Gromacs software, and the provided data were plotted, using Gnuplot (Williams & Kelley, 2015).

**Modeling of AGR2/TMED2 interactions**

The structure of the Golgi dynamics (GOLD) domain of human TMED2 (PDB ID: 5AZW; Nagae *et al*, 2016) and the NMR structure of the non-covalent dimer of AGR2 (PDB ID: 2LNS; Patel *et al*, 2013) were downloaded from the Brookhaven protein databank (Berman *et al*, 2000) followed by protein preparation in YASARA (Krieger & Vriend, 2014). Protein–protein docking was performed by using the web-server PatchDock (Duhovny *et al*, 2002; Schneidman-Duhovny *et al*, 2005). The AGR2 dimer or AGR2 monomer was defined as the receptor, and TMED2 was defined as the ligand. In each docking run, the top 500 decoys were subjected for refinement and rescoring in the web-server FireDock (Andrusier *et al*, 2007; Mashiach *et al*, 2008) retaining the top 10 scored decoys from each run. Analysis of interaction energies revealed that the AGR2 dimer provided significantly stronger interactions with TMED2 than the monomer. Visual inspection of the monomer and dimer AGR2 complexes with TMED2 and complementarity of their respective electrostatic surfaces were used to identify most likely interaction structures. Molecular images and electrostatic surfaces were generated using Chimera (Pettersen *et al*, 2004).

**Cryo-electron microscopy**

Vitrification of purified exosomes was performed using an automatic plunge freezer (EM GP, Leica) under controlled humidity and temperature (Dubochet & McDowall, 1981). The samples were deposited to glow-discharged electron microscope grids followed by blotting and vitrification by rapid freezing into and were observed using a 200 kV electron microscope (Tecnai $G^2$ T20 Sphera, FEI) equipped with a 4k × 4k CCD camera (model USC 4000, Gatan). Micrographs were acquired under low electron doses using the camera in binning mode 1 and at a nominal magnifications of 29,000×. Measurements were performed using the measuring tools available in Fiji (Schindelin *et al*, 2012).

**Expanded View** for this article is available online.

**Acknowledgements**
This work was funded by grants from INSERM, Institut National du Cancer (PLBIO INCa_5869), Région Bretagne, Rennes Métropole, Cancéropôle Grand Sud-Ouest to EC and EU H2020 MSCA ITN-675448 (TRAINERS) and MSCA RISE-734749 (INSPIRED) to AS, LAE and EC; the Swedish Research Council through grant 2014-3914 to LAE; Association François Aupetit (AFA), Université Diderot Paris 7, and the Investissements d'Avenir programme ANR-11-IDEX-0005-02, Sorbonne Paris Cité, Laboratoire d'excellence INFLAMEX to EOD; GACR 19-02014S to RH and PROMISE, 12CHN 204 Bilateral Greece-China Research Program of the Hellenic General Secretariat of Research and Technology and the Chinese Ministry of Research and Technology sponsored by the Program "Competitiveness and Entrepreneurship," Priority Health of the Peripheral Entrepreneurial Program of Attiki to AC. MM was supported by post-doctoral fellowships from FWO and INCa. JO was supported by a grant from Région Bretagne. Grants of supercomputing time were generously

provided by the Swedish National Infrastructure for Computing (SNIC). DYT is the Canada Research Chair in Molecular Genetics, and his work is supported by grants from the Canadian Institutes of Health Research.

## Author contributions

MM developed the ERMIT assay and performed the screen and all the subsequent validation analyses. OP, EP, AC performed the biostatistical and bioinformatic analyses. Y-PD, XT, DC-H, FDa, EO-D provided and analyzed human CD and UC samples. JH, LAE performed and analyzed the simulations of AGR2 dimers and AGR2-TMED2 protein–protein interactions. AD performed the cryoEM experiments. TA, JO, DF, FDe, LS performed experiments relative to the secretion of AGR2 and validated the impact of TMED2 expression alteration in cellular model systems. WH, M-CB, LJ-M, JT-M provided data generated in mice. DYT, GJ, TH, RH, MEF-Z, JT, AS provided reagents and worked the manuscript. DYT, GJ provided data obtained in *S. cerevisiae* with ER-MYTH. AC, EC, EOD conceived the study, performed analyses and wrote the manuscript.

## Conflict of interest

AC and EP are founders of e-NIOS (https://e-nios.com/), and AS, LAE, and EC are founders of Cell Stress Discoveries Ltd (https://cellstressdiscoveries.com/).

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
