## [Review Process File · EMBO Molecular Medicine]

Control of Anterior GRadiant 2 (AGR2) dimerization links endoplasmic reticulum proteostasis to inflammation

Marion Maurel, Joanna Obacz, Tony Avril, Yong-Ping Ding, Olga Papadodima, Xavier Treton, Fanny Danie, Eleftherios Pilalis, Johanna Hörberg, Wenyang Hou, Marie-Claude Beauchamp, Julien Tourneur-Marseille, Dominique Cazals-Hatem, Lucia Sommerova, Afshin Samali, Jan Tavernier, Roman Hrstka, Aurélien Dupont, Delphine Fessart, Frédéric Delom, Martin E. Fernandez-Zapico, Gregor Jansen, Leif A. Eriksson, David Y Thomas, Loydie Jerome-Majewska, Ted Hupp, Aristotelis Chatziioannou, Eric Chevet, Eric Ogier-Denis

Review timeline:

Submission date:	29 November 2018
Editorial Decision:	8 January 2019
Revision received:	1 March 2019
Editorial Decision:	22 March 2019
Revision received:	28 March 2019
Accepted:	7 April 2019

Editor: Céline Carret

Transaction Report:

1st Editorial Decision

8 January 2019

Thank you for the submission of your manuscript to EMBO Molecular Medicine. We have now heard back from the two referees whom we asked to evaluate your manuscript.

You will see from the comments pasted below that both referees are supportive of publication, only suggesting limited additions. I'd like to insist on referee 2's comment to identify the AGR2 secretory pathway in order to deepen the putatively targetable therapeutic signaling axes.

We would therefore welcome the submission of a revised version within three months for further consideration and would like to encourage you to address all the criticisms raised as suggested to improve conclusiveness and clarity. Please note that EMBO Molecular Medicine strongly supports a single round of revision and that, as acceptance or rejection of the manuscript will depend on another round of review, your responses should be as complete as possible.

I look forward to receiving your revised manuscript.

***** Reviewer's comments *****

Referee #1 (Remarks for Author):

This manuscript deals with the anterior gradient 2 (AGR2), a folding catalyst/chaperone that resides in the endoplasmic reticulum (ER) and has been implicated in ER proteostasis and associated with intestinal inflammation. The study provides molecular insights into the AGR2 mode of action with an emphasis on the regulation of AGR2 assembly into homodimers and also addresses the involvement of AGR2 and its dimers in inflammatory bowel disease (IBD) and Crohn's disease.

Major findings:

1. AGR2 forms stress-regulated homodimers in the ER:

- Molecular dynamics confirmed AGR2 homodimerization via E60 and verified the reduced dimer stability of the E60A mutant.
- AGR2 dimeric / monomeric equilibrium was investigated by molecular modeling.
- Chemical cross-linking combined with non-reducing and reducing electrophoresis followed by immunoblotting showed the predominance of AGR2 homodimers.
- The ERMIT assay, in which IRE1 luminal domain was replaced with various AGR2 constructs (wild-type, E60A, C81A, E60A/C81A double mutant), verified AGR2 homodimerization in the ER, while E60 key role in the homodimers formation was deduced from the dimerization defective mutants (E60A, E60A/C81A).
- Upon ER stress (induced by DTT, thapsigargin or tunicamycin), AGR2 homodimers dissociated in a dose-dependent manner, and 35S-methionine pulse-chase followed by AGR2 immunoprecipitation revealed ER stress-induced altered association of AGR2 with nascent proteins.

2. Identification AGR2 interactors as modulators of AGR2 homodimerization and characterization of TMED2 as a major enhancer of AGR2 dimerization:

- A specific ERMIT-based siRNA screen identified inhibitors and enhancers of AGR2 homodimerization. Their functional pathways implicated the dimeric AGR2 in protein folding and the monomeric form in managing ER stress.
- TMED2 was identified as a positive regulator that enhanced AGR2 homodimerization and co-immunoprecipitation revealed a TMED2-AGR2 complex that dissociated upon ER stress.
- AGR2 AA (K66A/Y111A), a mutant unable to interact with TMED2 (generated according to molecular modeling), failed to co-immunoprecipitate with TMED2.
- Protein-protein docking showed that TMED2 interaction with AGR2 monomers was unstable, while its interaction with AGR2 dimers rendered several conformers in which TMED2 simultaneously interacted with both AGR2 monomers.
- TMED2 overexpression resulted in enhanced AGR2 homodimerization, yet led to reduced AGR2 expression and reduced ERMIT signals due to AGR2 dwindling.
- TMED2 silencing led to enhanced AGR2 expression but decreased ERMIT signals, reflecting reduced homodimerization.

3. AGR2 dimerization had no effect on its chaperone activity but AGR2-TMED2 interplay regulated protein folding and trafficking and contributed to ER protein quality control:

- AGR2 regulation of cargo secretion was followed by ERMIT, monitoring the interaction of the constitutively monomeric AGR2 E60A with two plasma-membrane GPI-anchored proteins, CD59 and LYPD3.
- AGR2 contribution to the ER quality control and to protein secretion was followed with CD59 and its misfolded ER-retained C94S mutant, and both CD59 and C94S interacted similarly with either AGR2 or the TMED2 interaction-defective AGR2 AA.
- AGR2 silencing led to reduced expression of intracellular CD59 and C94S with no effect on their cell surface expression, suggesting a role of intracellular AGR2 in ER quality control.
- TMED2 silencing led to reduced expression of intracellular CD59 and C94S accompanied by similar effect on their cell surface expression.
- Overexpression of either AGR2 or AGR2 AA restored the total and cell surface expression of CD59 and C94S.
- While AGR2 silencing did not affect the secretion of alpha-1-antitrypsin under basal conditions, the ER retention of alpha-1-antitrypsin decreased in the absence of AGR2 upon ER stress.

- A role for AGR2 in ER proteostasis was deduced from the stabilized expression of MUC2 in the presence of AGR2.

4. Overexpression of TMED2 decreased AGR2 levels not via ERAD but through autophagy:

- Pharmacological inhibitors of the ERAD pathway did not recover the decreased levels of intracellular AGR2 induced by TMED2 overexpression.
- The decreased levels of intracellular AGR2 induced by TMED2 overexpression were recovered by chloroquin, implicating a lysosomal/autophagy-dependent degradation of AGR2.
- TMED2 overexpression lead to abnormal secretory features, including the release of aberrant AGR2 entities such as heterogeneous populations of extracellular vesicles.
- TMED2 silencing resulted in increased intracellular AGR2 and promoted elevated secretion of AGR2.
- The constitutively monomeric AGR2 E60A was secreted more efficiently than wild-type AGR2, while the constitutively dimeric AGR2 Δ 45 was retained inside the cell.
- The secretion of the TMED2 interaction-defective AGR2 AA was not affected by either overexpression or silencing of TMED2.

5. Pathophysiological relevance of AGR2 and its dimerization to intestinal inflammation in mice models and patients biopsies:

- In mice, alteration of TMED2 expression was related to AGR2-mediated monocytes attraction.
- Expression levels of AGR2, TMED2 and CD163 were altered in biopsies from IBD patients when compared to healthy controls.

This study is a nice example of the emerging link between ER proteostasis and the pathophysiology of various diseases. Proper folding of newly synthesized proteins in the ER is assisted by molecular chaperones and folding catalysts, whereas ER quality control and degradation mechanisms handle the burden of misfolded proteins. In this manuscript the authors address molecular mechanisms underlying the involvement of AGR2 in regulating ER protein quality control and the development of IBD. The major focus of this study is AGR2 homodimers: their regulation by assembly modulators, their response to ER stress and their effect on AGR2 function. To unveil AGR2 molecular functions and pathophysiological roles, the authors employed several complementary approaches with carefully designed controls, to provide clear and comprehensive data and straightforward conclusions. Most impressive is the development of ERMIT, an ER mammalian protein-protein interaction trap, which allowed them to detect AGR2 interactors. Using this elegant protein-protein interaction screen, the authors identified TMED2 as a major and specific enhancer of AGR2 dimerization and discovered that AGR2 homodimers were disrupted upon ER stress. They also demonstrated that variations in TMED2 expression resulted in AGR2 secretion and pro-inflammatory phenotypes, correlating deregulated levels of AGR2 dimerization modulators with IBD and Crohn's disease severity. The authors raise the possibility that AGR2 dimers act as sensors of ER homeostasis, which are disrupted upon ER stress and promote the secretion of AGR2 monomers, possibly representing systemic alarm signals for pro-inflammatory responses.

Minor points:

1. Indeed, in Fig. 3E, the levels of AGR2 co-immunoprecipitated with TMED2 are higher for the wild-type AGR2 (left panel) than for the TMED2 interaction-defective AGR2 AA mutant (right panel). However, in the input lower panels, wild-type AGR2 is hardly detected and no AGR2 AA is detected whatsoever. It confirms that TMED2 overexpression leads to decreased levels of AGR2, although independently of TMED2-AGR2 interaction (Fig. 3E,F), yet one wonders what is the "source" of the co-immunoprecipitated AGR2 species.

Referee #2 (Comments on Novelty/Model System for Author):

AGR2 perturbation in mammals influences disease processes including cancer progression and drug resistance, asthma, and inflammatory bowel disease. This article is a tour-de-force with data ranging from structural biology, biochemistry, proteomics, electron microscopy, mouse genetics to clinics to demonstrate the molecular mechanism of how AGR2 may participate to inflammation.

The novel results are that 1/modulation of AGR2 dimer formation by point mutations and expression of partner TMED2 (identified in this study) expression induces pro-inflammatory

phenotypes via autophagy-dependent processes or secretion of AGR2 and 2/the expression levels of AGR2 dimerization modulators including TMED2 are selectively deregulated in Crohn's disease.

The methods used are adequate and the results were well acquired and analysed and fairly discussed. The biochemical and cell biological data are particularly convincing.

Referee #2 (Remarks for Author):

This article is a tour-de-force with data ranging from structure biology to clinics demonstrating the role of AGR2 in inflammation. Indeed, AGR2 perturbation in mammals influences disease processes including cancer progression and drug resistance, asthma, and inflammatory bowel disease. The methods used are adequate and the results were well acquired and analysed and fairly discussed.

The one missing piece of data which would clarify the process described and potentially develop therapeutic strategy concerns the secretory pathway of AGR2: is it a conventional secretion ER-Golgi-PM or unconventional one from ER to PM or does it involve endosomes?

1st Revision - authors' response

1 March 2019

REVIEWER 1

This manuscript deals with the anterior gradient 2 (AGR2), a folding catalyst/chaperone that resides in the endoplasmic reticulum (ER) and has been implicated in ER proteostasis and associated with intestinal inflammation. The study provides molecular insights into the AGR2 mode of action with an emphasis on the regulation of AGR2 assembly into homodimers and also addresses the involvement of AGR2 and its dimers in inflammatory bowel disease (IBD) and Crohn's disease.

Major findings:

1. AGR2 forms stress-regulated homodimers in the ER:

- Molecular dynamics confirmed AGR2 homodimerization via E60 and verified the reduced dimer stability of the E60A mutant.
- AGR2 dimeric / monomeric equilibrium was investigated by molecular modeling.
- Chemical cross-linking combined with non-reducing and reducing electrophoresis followed by immunoblotting showed the predominance of AGR2 homodimers.
- The ERMIT assay, in which IRE1 luminal domain was replaced with various AGR2 constructs (wild-type, E60A, C81A, E60A/C81A double mutant), verified AGR2 homodimerization in the ER, while E60 key role in the homodimers formation was deduced from the dimerization defective mutants (E60A, E60A/C81A).
- Upon ER stress (induced by DTT, thapsigargin or tunicamycin), AGR2 homodimers dissociated in a dose-dependent manner, and 35S-methionine pulse-chase followed by AGR2 immunoprecipitation revealed ER stress-induced altered association of AGR2 with nascent proteins.

2. Identification AGR2 interactors as modulators of AGR2 homodimerization and characterization of TMED2 as a major enhancer of AGR2 dimerization:

- A specific ERMIT-based siRNA screen identified inhibitors and enhancers of AGR2 homodimerization. Their functional pathways implicated the dimeric AGR2 in protein folding and the monomeric form in managing ER stress.
- TMED2 was identified as a positive regulator that enhanced AGR2 homodimerization and co-immunoprecipitation revealed a TMED2-AGR2 complex that dissociated upon ER stress.
- AGR2 AA (K66A/Y111A), a mutant unable to interact with TMED2 (generated according to molecular modeling), failed to co-immunoprecipitate with TMED2.
- Protein-protein docking showed that TMED2 interaction with AGR2 monomers was unstable, while its interaction with AGR2 dimers rendered several conformers in which TMED2 simultaneously interacted with both AGR2 monomers.
- TMED2 overexpression resulted in enhanced AGR2 homodimerization, yet led to reduced AGR2 expression and reduced ERMIT signals due to AGR2 dwindling.
- TMED2 silencing led to enhanced AGR2 expression but decreased ERMIT signals, reflecting reduced homodimerization.

3. AGR2 dimerization had no effect on its chaperone activity but AGR2-TMED2 interplay regulated protein folding and trafficking and contributed to ER protein quality control:

- AGR2 regulation of cargo secretion was followed by ERMIT, monitoring the interaction of the constitutively monomeric AGR2 E60A with two plasma-membrane GPI-anchored proteins, CD59 and LYPD3.
- AGR2 contribution to the ER quality control and to protein secretion was followed with CD59 and its misfolded ER-retained C94S mutant, and both CD59 and C94S interacted similarly with either AGR2 or the TMED2 interaction-defective AGR2 AA.
- AGR2 silencing led to reduced expression of intracellular CD59 and C94S with no effect on their cell surface expression, suggesting a role of intracellular AGR2 in ER quality control.
- TMED2 silencing led to reduced expression of intracellular CD59 and C94S accompanied by similar effect on their cell surface expression.
- Overexpression of either AGR2 or AGR2 AA restored the total and cell surface expression of CD59 and C94S.
- While AGR2 silencing did not affect the secretion of alpha-1-antitrypsin under basal conditions, the ER retention of alpha-1-antitrypsin decreased in the absence of AGR2 upon ER stress.
- A role for AGR2 in ER proteostasis was deduced from the stabilized expression of MUC2 in the presence of AGR2.

4. Overexpression of TMED2 decreased AGR2 levels not via ERAD but through autophagy:

- Pharmacological inhibitors of the ERAD pathway did not recover the decreased levels of intracellular AGR2 induced by TMED2 overexpression.
- The decreased levels of intracellular AGR2 induced by TMED2 overexpression were recovered by chloroquin, implicating a lysosomal/autophagy-dependent degradation of AGR2.
- TMED2 overexpression lead to abnormal secretory features, including the release of aberrant AGR2 entities such as heterogeneous populations of extracellular vesicles.
- TMED2 silencing resulted in increased intracellular AGR2 and promoted elevated secretion of AGR2.
- The constitutively monomeric AGR2 E60A was secreted more efficiently than wild-type AGR2, while the constitutively dimeric AGR2 Δ 45 was retained inside the cell.
- The secretion of the TMED2 interaction-defective AGR2 AA was not affected by either overexpression or silencing of TMED2.

5. Pathophysiological relevance of AGR2 and its dimerization to intestinal inflammation in mice models and patients biopsies:

- In mice, alteration of TMED2 expression was related to AGR2-mediated monocytes attraction.
- Expression levels of AGR2, TMED2 and CD163 were altered in biopsies from IBD patients when compared to healthy controls.

We thank this reviewer for this very clear description of the main findings presented in our work

This study is a nice example of the emerging link between ER proteostasis and the pathophysiology of various diseases. Proper folding of newly synthesized proteins in the ER is assisted by molecular chaperones and folding catalysts, whereas ER quality control and degradation mechanisms handle the burden of misfolded proteins. In this manuscript the authors address molecular mechanisms underlying the involvement of AGR2 in regulating ER protein quality control and the development of IBD. The major focus of this study is AGR2 homodimers: their regulation by assembly modulators, their response to ER stress and their effect on AGR2 function. To unveil AGR2 molecular functions and pathophysiological roles, the authors employed several complementary approaches with carefully designed controls, to provide clear and comprehensive data and straightforward conclusions. Most impressive is the development of ERMIT, an ER mammalian protein-protein interaction trap, which allowed them to detect AGR2 interactors. Using this elegant protein-protein interaction screen, the authors identified TMED2 as a major and specific enhancer of AGR2 dimerization and discovered that AGR2 homodimers were disrupted upon ER stress. They also demonstrated that variations in TMED2 expression resulted in AGR2 secretion and pro-inflammatory phenotypes, correlating deregulated levels of AGR2 dimerization modulators with IBD and Crohn's disease severity. The authors raise the possibility that AGR2 dimers act as sensors of ER homeostasis, which are disrupted upon ER stress and promote the secretion of AGR2 monomers, possibly representing systemic alarm signals for pro-inflammatory responses.

We thank this reviewer for the nice comments on our work

Reviewer#1 - remarks breakdown

Comment R1-1: Indeed, in Fig. 3E, the levels of AGR2 co-immunoprecipitated with TMED2 are higher for the wild-type AGR2 (left panel) than for the TMED2 interaction-defective AGR2 AA mutant (right panel). However, in the input lower panels, wild-type AGR2 is hardly detected and no AGR2 AA is detected whatsoever. It confirms that TMED2 overexpression leads to decreased levels of AGR2, although independently of TMED2-AGR2 interaction (Fig. 3E,F), yet one wonders what is the "source" of the co-immunoprecipitated AGR2 species.

Response R1-1: As indicated by reviewer#1, TMED2 overexpression leads to decrease AGR2_{wt} expression levels (Figure 3E provided in Figure R1A, Figure 3F, and Figure R1B). We observed also a decreased expression of AGR2_{AA} upon TMED2 overexpression (Figure R1A and Figure R1C) although its AGR2 mutant should not be able to interact with TMED2 according to our *in silico* structural analyses of protein/protein interactions. Over-exposed images of Figure 3E discussed by the reviewer#1 are shown in Figure R1B and R1C, and confirm the presence of reduced amounts of AGR2_{wt} and AGR2_{AA} in TMED2 over-expressing cells. The source of AGR2 observed in co-immunoprecipitation experiments is therefore provided by the cells. As suggested by the reviewer#1, this underlines the possibility that TMED2 regulates AGR2 expression independently to direct TMED2/AGR2 interactions. This might occur through mechanisms associated with some unconventional release of AGR2 in the extracellular space (see below). A sentence has been added to address this point on p9 of the revised manuscript.

Figure R#1_1

R#1_1A

R#1_1B

R#1_1C

Figure R#1-1: Validation of AGR2/TMED2 interaction. Co-immunoprecipitation of AGR2 wild-type (wt, left panel) or AGR2 AA mutant (right panel) with TMED2 in HEK293T cells. Immunoprecipitation was performed with mouse anti-Flag antibody to pull-down the ectopic protein tag (R#1_1A). Longer exposure of the western blots framed in red boxes (1) and (2) were provided in R#1_1B and R#1_1C respectively.

REVIEWER 2 (Comments on Novelty/Model System for Author):

AGR2 perturbation in mammals influences disease processes including cancer progression and drug resistance, asthma, and inflammatory bowel disease. This article is a tour-de-force with data ranging from structural biology, biochemistry, proteomics, electron microscopy, mouse genetics to clinics to demonstrate the molecular mechanism of how AGR2 may participate to inflammation.

The novel results are that 1/modulation of AGR2 dimer formation by point mutations and expression of partner TMED2 (identified in this study) expression induces pro-inflammatory phenotypes via autophagy-dependent processes or secretion of AGR2 and 2/the expression levels of AGR2 dimerization modulators including TMED2 are selectively deregulated in Crohn's disease.

The methods used are adequate and the results were well acquired and analysed and fairly discussed. The biochemical and cell biological data are particularly convincing.

Remarks for Author:

This article is a tour-de-force with data ranging from structure biology to clinics demonstrating the role of AGR2 in inflammation. Indeed, AGR2 perturbation in mammals influences disease processes including cancer progression and drug resistance, asthma, and inflammatory bowel disease. The methods used are adequate and the results were well acquired and analysed and fairly discussed.

We thank this reviewer for this very kind comments on our work.

Reviewer#2 - remarks breakdown

Comment R2-1: The one missing piece of data which would clarify the process described and potentially develop therapeutic strategy concerns the secretory pathway of AGR2: is it a conventional secretion ER-Golgi-PM or unconventional one from ER to PM or does it involve endosomes?

Response R2-1: We thank the reviewer#2 for this important remark. To test if AGR2 secretion is dependent of the endo-lysosomal pathway, we have performed additional experiments in which AGR2 was overexpressed in HEK293T cells which were treated or not with brefeldin A (BFA) and/or bafilomycin A. BFA inhibits secretory and transmembrane protein transport from the ER to the Golgi apparatus by blocking a membrane-associated ARF-GEF activity (sec7 domain containing proteins) and bafilomycin A is an inhibitor of the endosomal V-ATPase and prevents the entry of protons in the endosome, hence its acidification. Total and secreted AGR2 levels were analysed by western blot.

Figure R#2_1

Figure R#2-1: Cellular mechanisms of AGR2 secretion. HEK293T cells were transfected with AGR2_{wt} and were treated with brefeldin A (BFA, 5 µg/ml), bafilomycin A1 (baf, 200 nM), or both (B+B). Western blot analysis of intracellular and secreted AGR2 upon 2 and 4 hours treatment were shown. eAGR2, extracellular AGR2; AGR2, intracellular AGR2. Actin (ACT) served as a loading control.

As shown in Figure R#2_1, treatments with drugs were carried out for relatively short time periods to minimize the impact on the ER stress response. BFA treatment led to reduced intracellular AGR2 expression as well as the appearance of a lower mobility form that might correspond to absence of peptide signal cleavage as previously shown (Genes Chromosomes Cancer. 2005 Jul;43(3):249-59) which was observed after 2 to 4 hours of treatment. In contrast, no change was observed upon treatment with bafilomycin. As expected BFA treatment reduced AGR2 secretion after 2 hours, but led to further release of AGR2 after 4h, suggesting a bypass of the conventional secretory pathway. Again bafilomycin had no effect on AGR2 secretion. Interestingly the secretion of AGR2 was abrogated upon treatment with both BFA and bafilomycin thus suggesting that the secretion of AGR2 might be in part dependent on the endo-lysosomal system as also observed upon overexpression of TMED2 in Fig 5 of the manuscript. We have added this figure in Supplemental material (Fig S5D) and a corresponding sentence on p12 of the revised manuscript.

2nd Editorial Decision

22 March 2019

Thank you for the submission of your revised manuscript to EMBO Molecular Medicine. We have now received the enclosed reports from the referees that were asked to re-assess it. As you will see the reviewers are now globally supportive and I am pleased to inform you that we will be able to accept your manuscript pending minor editorial amendments.

Please submit your revised manuscript within two weeks. I look forward to seeing a revised form of your manuscript as soon as possible.

***** Reviewer's comments *****

Referee #1 (Remarks for Author):

The authors have adequately responded to the comments raised by both reviewers and revised the manuscript accordingly.

Referee #2 (Remarks for Author):

The authors have satisfactorily answered the reviewers' requests.

2nd Revision - authors' response

28 March 2019

Authors made the requested editorial changes.

Corresponding Author Name: Eric Chevet
Journal Submitted to: EMBO MOLECULAR MEDICINE
Manuscript Number: EMM-2018-10120